# Suppression of mechanical hypersensitivity and change in the expression of the dopamine D2 receptor by administration of anti-CGRP antibody into the trigeminal ganglion in trigeminal neuropathic pain model rats

Hiroharu Maegawa[ID]*, Nayuka Usami, Chiho Kudo, Hitoshi Niwa

Department of Dental Anesthesiology, The University of Osaka Graduate School of Dentistry, Suita, Osaka, Japan

* maegawa.hiroharu.dent@osaka-u.ac.jp

## Abstract

Calcitonin gene-related peptide (CGRP) and dopamine D2 receptor (D2 receptor) are associated with neuropathic pain. However, their relationship is not well understood. To better establish their relationship in trigeminal neuropathic pain, we examined male rats with infraorbital nerve (ION) ligation. Rats with ION ligation were administered CGRP or anti-CGRP antibody into the trigeminal ganglion (TG). The change in the head-withdrawal threshold was measured using von Frey filament. Immunohistochemical staining for phosphorylated extracellular signal-regulated kinase (pERK) was also performed in the trigeminal spinal subnucleus caudalis (Vc). CGRP was detected in the TG and Vc by immunohistochemical staining, and glial fibrillary acidic protein (GFAP) in the TG and dopamine D2 receptor in the Vc were detected in the same way. Antibody administration restored the head-withdrawal threshold to mechanical stimuli, which had decreased after ION ligation. Furthermore, the number of pERK-immunoreactive (-IR) neurons in the Vc, which increased following ION ligation, declined. The ratio of CGRP-IR TG neurons, large-sized CGRP-IR TG neurons, and TG neurons encircled with the GFAP-IR cells increased after ION ligation and decreased after antibody administration. Moreover, the immunoreactivity of CGRP and D2 receptor in the Vc increased after ION ligation and decreased after antibody administration. There were no significant differences in the head-withdrawal threshold, pERK-IR cell count, ratio of CGRP-IR TG neurons, ratio of size of CGRP-IR TG neurons, ratio of TG neurons encircled with GFAP-IR cells, and immunoreactivity of CGRP and D2 receptor in the Vc between the ION-ligated rats with and without CGRP. These findings suggest that the administration of an anti-CGRP antibody into the TG is involved in the suppression of trigeminal neuropathic pain and the D2 receptor expression in the Vc.

**Data availability statement:** All relevant data are within the manuscript and its Supporting Information files.

**Funding:** This research was funded by JSPS KAKENHI, grant number 22K10167. The funder had no role in study design, data collection and analysis, decision to publish, or preparation of the manuscript.

**Competing interests:** The authors have declared that no competing interests exist.

## Introduction

Calcitonin gene-related peptide (CGRP) is associated with neuropathic pain. In the spinal dorsal horn, rats with sciatic nerve ligation showed an increase in immunoreactivity of CGRP and CGRP protein [1]. Increased CGRP levels were also detected in the spinal cord of a cisplatin-induced neuropathic pain mouse model [2]. In a rat model of lingual neuropathic pain, lower mechanical and heat thresholds to tongue stimulation were restored after the administration of a CGRP receptor antagonist into the trigeminal ganglion (TG) [3]. In rats with infraorbital nerve (ION) ligation, mechanical allodynia was reduced by the intraperitoneal (i.p.) administration of a CGRP receptor blocker [4], whereas the mechanical and heat hypersensitivities were reduced by the intracerebroventricular administration of an anti-CGRP antibody [5]. Mechanical hypersensitivity was also reduced in a mouse model of cisplatin-induced neuropathic pain by the subcutaneous administration of an anti-CGRP antibody [2].

TG neurons produce CGRP, and CGRP is then released from their nerve endings [6]. TG neurons and satellite glial cells have CGRP receptors [7]. CGRP is also released from the cell bodies of TG neurons [8], and it activates the satellite glial cells in the TG [9]. The CGRP receptor binds to the Gs protein, activating adenylate cyclase (AC), causing an increase in cyclic adenosine monophosphate (cAMP) and protein kinase A (PKA) activation [10]. The signal transduction mechanism via the CGRP receptor is thought to be involved in neuropathic pain.

Pain control mechanisms derived from the dopaminergic nervous system, which also apply to neuropathic pain, are known to exist. Previous studies have found that administering a dopamine D2 receptor (D2 receptor) agonist reduced mechanical hypersensitivity in rats with infraorbital nerve (ION) ligation [11], as well as mechanical and heat hypersensitivities in rats with sciatic nerve ligation [12]. The hypothalamic A11 nucleus, which is among the dopaminergic nuclei, modulates neuropathic pain [11,13]. The spinal dorsal horn and trigeminal spinal subnucleus caudalis (Vc) are known to contain dopamine receptors [13,14]. The D2 receptor expression is increased in the spinal dorsal horn following the spinal nerve ligation [14,15]. The immunoreactivities of the CGRP and D2 receptor increased in the Vc following ION ligation and decreased after the intracerebroventricular administration of an anti-CGRP antibody [16]. In this manner, the dopaminergic nervous system is linked to neuropathic pain. The expression of dopamine receptors can also modulate neuropathic pain, which has been linked to CGRP because CGRP can affect the protein expression via intracellular transduction mechanisms after binding to its receptor [10]. However, the relationship between CGRP and the dopaminergic nervous system in neuropathic pain remains poorly understood. We have previously reported that the anti-CGRP antibody was involved in the attenuation of trigeminal neuropathic pain and change in D2 receptor immunoreactivity in the Vc [5,16], but the area where it acted was not clear due to the intracerebroventricular administration of the antibody.

The present study aimed to further elucidate the relationship between CGRP and the dopaminergic nervous system in neuropathic pain, especially, the site where the anti-CGRP antibody controls neuropathic pain and affects D2 receptor

immunoreactivity. We hypothesized that the site where the anti-CGRP antibody controls neuropathic pain is the TG. Therefore, we administered anti-CGRP antibodies to the TG of a rat model of trigeminal neuropathic pain to determine whether neuropathic pain could be attenuated or whether the changes in D2 receptor immunoreactivity in the Vc occurred.

## Materials and methods

The protocol of the current study was approved by the Animal Experimentation Committee of The University of Osaka Graduate School of Dentistry (R4-003). The experiments were conducted following the guidelines stipulated in the International Association for the Study of Pain [17].

### Animal model of trigeminal neuropathic pain by ION ligation

Seven-week old (weight: 170–200 g) male Wister rats (Japan SLC, Hamamatsu, Japan) were used in the present study. They were raised on a 12-h dark/light cycle and had access to water and food at all times. ION ligation was conducted as described previously [11]. The rats were anesthetized with an i.p. administration of a saline solution containing midazolam (2.0 mg/kg, Sandoz, Tokyo, Japan), medetomidine (0.375 mg/kg, Zenoaq, Fukushima, Japan), and butorphanol (2.5 mg/kg, Meiji Seika Pharma, Tokyo, Japan). This anesthetic was used in all experiments in the present study. Rats were placed in the supine position, and an intraoral incision was performed along the gingiva–buccal margin proximal to the first molar [18]. Then, the left ION was exposed using blunt dissection, and two ligatures (4–0 silk) were tied loosely around it. The sham operation was performed in the same way but without nerve ligation.

### Assessment of reactivity to mechanical stimulation

Mechanical stimuli using a von Frey filament (North Coast Medical, Morgan Hill, CA, USA) were applied to the rats' left maxillary whisker pad skin to assess the reactivity to mechanical stimulation. The filaments used for mechanical stimulation were 0.4, 0.6, 1, 2, 4, 6, 8, 10, 15, 26, 35, and 60 g. The force applied with the filament to the rats was limited to 60 g to prevent tissue damage [19]. Rats were placed in a plastic box with a small hole in the front wall, and their snouts protruded through the hole. Even before the nerve ligation, rats were trained daily to keep their snouts protruding from the hole in the box. When rats were subjected to mechanical stimulation, they were not restrained and were free to escape from the stimuli. The minimum force that elicits an escape response three out of five times or more was identified as the head-withdrawal threshold. The head-withdrawal threshold was measured before the nerve ligation and at 3, 7, 14, and 21 days after the nerve ligation. The time course of the head-withdrawal threshold was analyzed using Friedmann's test followed by Wilcoxon's signed rank sum test as a multiple comparison to compare the threshold within the group, and a Mann–Whitney test was used for a comparison of data obtained at the same time point between the sham-operated and nerve-ligated rats. In the ION-ligated rats (n = 6), significant decreases in head-withdrawal thresholds at 7, 14, and 21 days were observed post-ligation as compared to the threshold pre-ligation ($p < 0.05$ each, S1 File in S1 Data). The sham-operated rats (n = 6) showed no significant difference in head-withdrawal thresholds before and after the sham operation. The ION-ligated rats had significantly lower head-withdrawal thresholds at 3, 7, 14, and 21 days after the ligation than the sham-operated rats at the same time ($p < 0.01$ each). In short, the ION-ligated rats demonstrated a reduction in head-withdrawal thresholds post-ligation, lasting for 21 days, which was consistent with the results of our previous studies [5,11]. Based on these findings, the ION-ligated rats with reduced head-withdrawal thresholds (< 8 g at 7 days after nerve ligation) were used in the subsequent experiments as rats with chronic constriction injury of the ION (ION-CCI).

### Implantation of a stainless-steel cannula for drug administration into the TG

After measuring the head-withdrawal threshold at 7 days after nerve ligation, the rats were anesthetized and a metal cannula was implanted on their skull for drug administration into the TG. The experimental schedule is shown in Fig 1A. The rats were anesthetized and placed on a stereotaxic apparatus (Narishige, Japan), with the incisor bar placed 3.3 mm

A

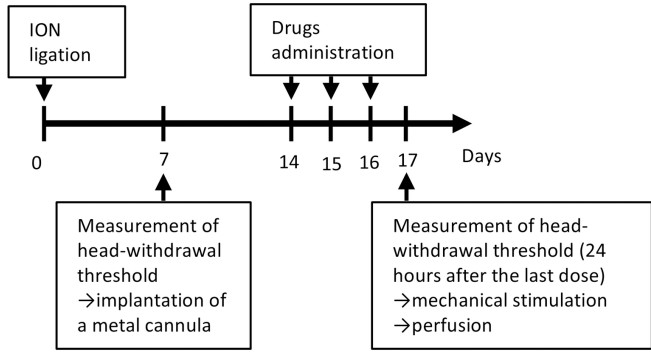

B

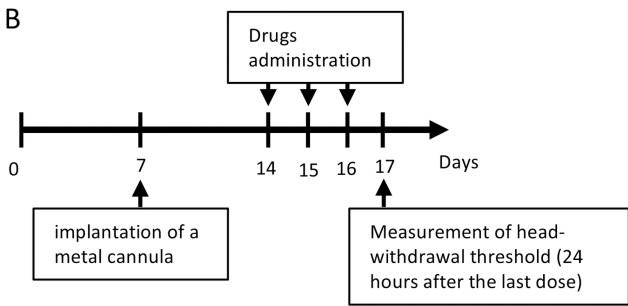

C

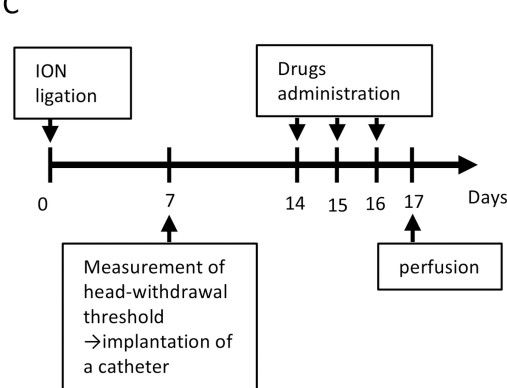

**Fig 1. The experimental schedules.** (A) the day of infraorbital nerve (ION) ligation was considered as day 0. At 7 days post-ligation, a nerve-ligated rat with a decrease in the head-withdrawal threshold (a rat with chronic constriction injury of the ION: ION-CCI rat) underwent a metal cannula implantation. Drugs were administered at 7, 8, and 9 days after the cannula implantation. The head-withdrawal threshold was measured at 24 hours after the last dose; then, mechanical stimulation was applied at the rats' whisker pad skin, and they were perfused after 5 minutes. (B) When naïve rats reached 8 weeks old, a metal cannula was implanted in the same manner described in panel (A). Drug administration was performed in the same manner described in panel (A). The head-withdrawal threshold was measured at 24 hours after the last dose. (C) ION ligation was performed in the same manner described in panel (A). At 7 days after the ligation, a catheter was implanted to an ION-CCI rat for the intracisternal administration of drugs. Drug administration was performed in the same manner described in panel (A). The ION-CCI rats were perfused at 24 hours after the last dose.

below the ear bars. A small hole was created using a dental drill. A 26-gauge stainless steel cannula (guide cannula) (Bio Research Center, Aichi, Japan) was inserted (stereotaxic coordinates: 2.8 mm anterior from lambda, 2.7 mm lateral to the midline, and 9 mm beneath the skull surface). To anchor the guide cannula, stainless-steel screws were placed in the skull

around the cannula. Then, the cannula and screws were fixed with dental acrylic. Then, a dummy cannula was inserted into the guide cannula to ensure patency until drug administration. After the cannula implantation, no significant change in the head-withdrawal threshold was observed.

Additionally, when the naïve rats reached the age of 8 weeks, the metal cannula was placed in the same manner as described above. The experimental schedule is shown in Fig 1B.

### Drug administration into the TG

At 7 days after the metal cannula implantation, drug administration began. Drugs were given to the TG once a day for 3 days. The drug was administered using an internal cannula, a 1-µL Hamilton syringe, and a polyethylene tube connecting the cannula and syringe. A dummy cannula was removed and replaced with a 33-gauge internal cannula. The internal cannula had a 1–mm projection from the guide cannula tip. The following drugs were used: CGRP (0.2 pmol/0.5 µL/day, C0292, Sigma-Aldrich, St. Louis, MO, USA); vehicle (saline, 0.5 µL/day); anti-CGRP antibody (0.5 µg/0.5 µL/day, C7113, Sigma-Aldrich, St. Louis, MO, USA), and control immunogloblin G (IgG; 0.5 µg/0.5 µL/day, ab18443, Abcam, Cambridge, UK). The experimental groups were as follows: sham-operated rats (sham), ION-CCI rats (ION-CCI), ION-CCI rats receiving the vehicle (ION-CCI + vehi), ION-CCI rats receiving CGRP (ION-CCI + CGRP), ION-CCI rats receiving control IgG (ION-CCI + ctrl IgG), and ION-CCI rats receiving the anti-CGRP antibody (ION-CCI + CGRP Ab). Before administering the anti-CGRP antibody and ctrl IgG, sodium azide was removed from the antibodies using an antibody purification kit (CSR-APK-10A, Cosmo Bio, Co. Ltd., Tokyo, Japan). The drug was administered into the TG slowly for more than a minute. When the drug administration into the TG was completed, the internal cannula was removed and a dummy cannula was inserted into the guide cannula again. On the day following the last drug administration (24 hours after the last dose), the head-withdrawal threshold was measured. Additionally, because the ION-CCI + CGRP Ab rats showed a change in the head-withdrawal thresholds, the time course of the threshold was examined in the group. At 7 days after the metal cannula implantation to naïve rats, they also received CGRP administration through the metal cannula in the same manner, and the head-withdrawal thresholds were measured at 24 hours after the last dose. The experimental groups were as follows: naïve rats receiving the vehicle (naïve + vehi), and naïve rats receiving CGRP (naïve + CGRP). The investigator who measured the head-withdrawal threshold was not informed of the experimental procedure that the rats underwent.

### Catheter placement for intracisternal drug administration

An intracisternal catheter was inserted into the subarachnoid space according to the method described previously [20]. After the ION-CCI rats were anesthetized, the head hair was shaved and the skull was exposed. A small hole was drilled using dental drill (coordinates: 5 mm caudal to the lambda and 2 mm left to the midline). The intracisternal catheter was a silicone tube (5 cm in length, 0.5 mm in diameter; Eastsidemed Inc., Tokyo, Japan) filled with saline, and inserted through the hole. The catheter tip was placed at the cisterna magna. The length of a silicone tube was 6 mm from the hole to the tip. After catheter placement, cerebrospinal fluid reflux was confirmed. To anchor the catheter, stainless-steel screws were placed in the skull around the catheter. Then, the catheter and the screws were fixed with dental acrylic. A plastic fiber was inserted into the opposite end of the catheter to seal it. At 2 days postoperatively, saline (10 µL) was poured into the catheter to maintain the patency. At 7–9 days postoperatively, the drugs were administered though the catheter once a day (vehicle, CGRP, ctrl IgG, and anti-CGRP antibody). The dosage of each drug is as follows: CGRP, 4 pmol/10 µL; vehicle, 10 µL; ctrl IgG, 10 µg/10 µL; and anti-CGRP antibody, 10 µg/10 µL. Following the drug administration, the catheter was flushed by saline (10 µL) and filled by it. The experimental schedule is shown in Fig 1C.

Immunohistochemistry for CGRP, D2 receptor, phosphorylated extracellular signal-regulated kinase (pERK), neuronal nuclei (NeuN), receptor activity modifying protein 1 (RAMP1), and glial fibrillary acidic protein (GFAP).

After administering the drug into the TG and measuring the head-withdrawal threshold, the rats were anesthetized. Then, mechanical stimuli (15 g, I Hz, 5 min) were applied to the skin of the left maxillary whisker pad in rats using a von

Frey filament. At 5 minutes after stimulation, the rats were perfused transcardially with 150 mL of phosphate-buffered saline (PBS, pH 7.4), followed by 300 mL of PBS containing 4% paraformaldehyde. Following perfusion, the dye was administered via a stainless-steel cannula. Then, the brain was removed. Following brain removal, dye diffusion into the TG in the ipsilateral side to the ION ligation was confirmed, and the TG in the ipsilateral side to the ION ligation was removed. The removed brains and TGs were post-fixed overnight in the perfusate before being soaked in 30% sucrose in 0.1-M phosphate buffer. Serial transverse sections of the Vc (50-μm thickness) and sections of the TG (15-μm thickness) in the horizontal plane along the long axis were made with a freezing microtome (Yamato Kohki Industrial Co. Ltd., Saitama, Japan). Every third section of TG and Vc was used for immunohistochemical staining. Additionally, ION-CCI rats receiving intracisternal drug administration were also perfused 24 hours after the last dose in the same manner described above. However, no mechanical stimulation was performed on these rats.

For pERK immunostaining with diaminobenzidine (DAB), the Vc sections were incubated with 0.3% hydrogen peroxide in methanol for 20 min, followed with 1% normal goat serum (S-1000, Vector Labs, Burlingame, CA, USA) and 0.3% Triton X-100 for 1 h, and finally with an anti-pERK antibody (4376, 1:1000, Cell Signaling Technology, MA, USA) overnight. Subsequently, the sections were incubated with a biotinylated anti-rabbit antibody (BA-1000, 1:200, Vector Labs) for 1 h, followed by an avidin–biotin–peroxidase complex (PK-6100, Vector Labs). Then, the sections were reacted with DAB tetrahydrochloride, ammonium nickel sulfate, hydrogen peroxide, and Tris–HCl buffer (SK-4100, DAB-kit, Vector Labs, CA, USA) to visualize the products of the previous reaction. Then, the sections were arranged on gelatin-coated glass slides and cover-slipped after air drying.

The following molecules were targeted for immunofluorescent staining: CGRP, D2 receptor, pERK, NeuN, RAMP1, and GFAP. Since CGRP receptor consists RAMP1 and calcitonin receptor-like receptor (CLR), RAMP1 is a component of CGRP receptor [10]. For immunofluorescent staining, the sections of TG and Vc were incubated with 0.3% hydrogen peroxide in methanol for 20 min, followed with 10% normal goat serum (S-1000, Vector Labs, Burlingame, CA, USA) and 0.3% Triton X-100 for 1 h. Then, the sections were incubated with primary antibodies, 0.3% Triton X-100, and 2% normal goat serum overnight. The antibodies used in the present study are shown in Table 1. The subsequent experiments were conducted in a dark environment. The sections were incubated with AlexaFluor 488-conjugated goat anti-mouse IgG (A32723), AlexaFluor 555-conjugated goat anti-rabbit IgG (A32732), and 2% normal goat serum for 120 min. The sections were processed in the same way as described above. Double staining was performed in the following combinations: CGRP and D2 receptor (324393); RAMP 1 and D2 receptor (sc-5303); pERK and NeuN; D2 receptor (324393) and NeuN; and D2 receptor (sc-5303) and pERK.

The pERK-immunoreactive (-IR) cells visualized with DAB were observed in the Vc using an optical microscope (Olympus, BX51, Tokyo, Japan). As reported previously [5,11,16], pERK-IR cells were primarily found in the superficial layers of the Vc in the ipsilateral side to the ION ligation. The number of pERK-IR cells was counted in every third section across

**Table 1. Antibodies used in immunofluorescence staining.**

| Antibody | Manufacturer | Product Number | Host | Dilution |
|---|---|---|---|---|
| CGRP | Sigma-Aldrich, MO, USA | c7113 | Mouse | 1:400 |
| D2 receptor | Calbiochem, Darmstadt, Germany | 324393 | Rabbit | 1:2500 |
| D2 receptor | Santa Cruz Biotechnology, TX, USA | sc-5303 | Mouse | 1:100 |
| GFAP | Sigma-Aldrich, MO, USA | G3893 | Mouse | 1:400 |
| NeuN | Cell Signaling Technology, MA, USA | 94403 | Mouse | 1:1500 |
| pERK | Cell Signaling Technology, MA, USA | 4376S | Rabbit | 1:200 |
| RAMP1 | Proteintech, IL, USA | 10327-1-AP | Rabbit | 1:50 |
| AlexaFluor 488 goat anti-mouse IgG | Thermo Fisher Scientific IL, USA | A32723 | Goat | 1:500 |
| AlexaFluor 555 goat anti-rabbit IgG | Thermo Fisher Scientific IL, USA | A32732 | Goat | 1:500 |

the entire Vc. Five consecutive sections were chosen at approximately 1400 mm caudal from the obex because they contained the most pERK-IR cells. Then, the average number of the pERK-IR cells in each section of the ipsilateral side to the ION ligation was calculated.

A confocal laser scanning microscope (LSM700, Carl-Zeiss, Tokyo, Japan) was used to examine fluorescently stained sections. Immunofluorescence images were captioned using a fluorescent microscope system (LSM software ZEN 2009, Carl-Zeiss, Tokyo, Japan). Three TG sections were selected randomly as described previously [3]. The ratio of CGRP-IR neurons in the ophthalmic (V1)/maxillary (V2) branch regions was calculated as follows: the number of CGRP-IR TG neurons/the total number of TG neurons × 100. Additionally, the area of the CGRP-IR TG neurons was measured using ImageJ (NIH, Bethesda, MD, USA), which was then classified into small (< 400 μm²), medium (400–800 μm²), and large (> 800 μm²) [3]. Moreover, regarding the GFAP-IR cells, the number of TG neurons encircled with GFAP-IR cells in over two-thirds of the cell circumference was counted according to a method described previously [3]. Next, the entire extent of the Vc was observed from the rostral to the caudal side based on the immunostaining products of CGRP and D2 receptor. The immunostaining products of CGRP and D2 receptor were observed throughout the region, with no obvious bias noted. Four sections were selected to assess these parameters from each rat (0, 800, 1600, and 2400 mm from the obex). The area of the luminescent site in the V2 branch region of the Vc was calculated with ImageJ's auto threshold function. The ratio of the luminescent sites was calculated as follows: area of luminescent sites/area of the field of view × 100 [11,16]. The proportion was classified as immunoreactivity. Then, the mean immunoreactivities of CGRP and D2 receptor in the Vc were calculated in the ipsilateral side to ION ligation. The immunofluorescent images were taken at a site approximately 1400 mm caudal from the obex, as in the case of staining with DAB. The investigator who performed the immunohistochemical analyses was not informed of the experimental procedures.

## Statistical analysis

Box and whisker plots were used to display behavioral data. For the CGRP-IR TG neurons, the percentage of the CGRP-IR TG neurons divided by area was shown. All the other data were presented as mean ± standard error of the mean. Statistical analyses were performed using R (version 4.3.2, The R Project for Statistical Computing, www.r-project.org). Shapiro–Wilk test was used to test for normality. The head-withdrawal thresholds after administering CGRP or anti-CGRP antibody into the TG of the ION-CCI rats were analyzed using a Kruskal–Wallis test followed by a Steel–Dwass post hoc test for multiple comparisons. The time course of the head-withdrawal threshold of the ION-CCI + CGRP Ab rats was analyzed using a Friedmann's test followed by Wilcoxon's signed rank sum test as a multiple comparison. The head-withdrawal thresholds of the naïve rats receiving CGRP were analyzed using a Friedmann's test followed by Wilcoxon's signed rank sum test as a multiple comparison to compare the thresholds within the group, and a Mann–Whitney test was used for the comparison of data obtained at the same time point between the naïve rats receiving a vehicle and naïve rats receiving CGRP. The ratio of CGRP-IR TG neurons divided by area to the total number of CGRP-IR TG neurons was analyzed using a chi-square test with Bonferroni correction. All the other results (number of pERK-IR cells in the Vc, ratio of CGRP-IR TG neurons, CGRP immunoreactivity in the Vc, and immunoreactivity of D2 receptor in the Vc) were evaluated by performing one-way analysis of variance (ANOVA) and a Tukey's post hoc test for multiple comparisons. Significance was determined at $p < 0.05$.

## Results

### Reactivities to mechanical stimulation following drug administration into the TG

The Kruskal–Wallis test indicated a significant difference in the head-withdrawal threshold ($p < 0.001$, n = 6 in each group, Fig 2A). Post hoc tests revealed marked reductions in the threshold for the ION-CCI, ION-CCI + vehi, ION-CCI + CGRP, and ION-CCI + ctrl IgG groups as compared to the sham group ($p < 0.05$ each, Fig 2A, S1 File in S1 Data). A significant

increase in the threshold was found in the ION-CCI + CGRP Ab group as compared to those in the ION-CCI, ION-CCI + vehi, ION-CCI + CGRP, and ION-CCI + ctrl IgG groups ($p < 0.05$ each, Fig 2A). The analysis of the time course of the head-withdrawal threshold of the ION-CCI + CGRP Ab group showed a significant difference ($p < 0.001$, n = 6, Fig 2B). Significant increases in the head-withdrawal threshold were noted at 1 and 2 days after the antibody administration as compared with that before its administration ($p < 0.05$ each).

Most of the pERK-IR cells were located in the superficial layers of the V2 branch region of the Vc. Photomicrographs of the pERK-IR cells are shown in Fig 3A (S1-S6 Figs in S1 Data), and photomicrographs of these cells with high magnification are shown in Fig 3B (S7-S12 Figs in S1 Data). A marked main effect was found (F [5, 174] = 33.641, $p < 0.001$, n = 6

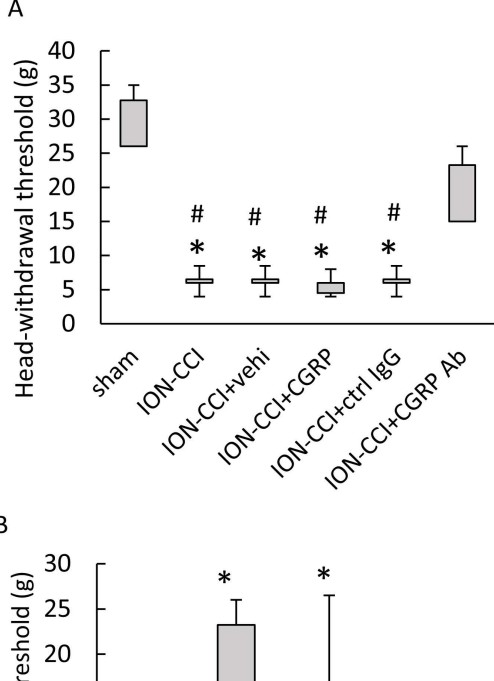

**Fig 2. Behavioral reactivities to mechanical stimulation after administering drugs into the trigeminal ganglion (TG).** (A) The head-withdrawal threshold to mechanical stimulation following drug administration. Data were analyzed using the Kruskal–Wallis test followed by the Steel–Dwass test. N = 6 in each group. Marked reductions in the threshold for the ION-CCI, ION-CCI + vehi, ION-CCI + CGRP, and ION-CCI + ctrl IgG groups as compared to the sham group were observed. A significant increase in the threshold was found in the ION-CCI + CGRP Ab group as compared to the ION-CCI, ION-CCI + vehi, ION-CCI + CGRP, and ION-CCI + ctrl IgG groups. * $p < 0.05$ compared to the sham group. # $p < 0.05$ compared to the ION-CCI + CGRP group. Sham: sham-operated rats; ION-CCI: rats with chronic constriction injury of the infraorbital nerve; ION-CCI + vehi: ION-CCI rats receiving a vehicle; ION-CCI + CGRP: ION-CCI rats receiving calcitonin gene-related peptide; ION-CCI + ctrl IgG: ION-CCI rats receiving control IgG; ION-CCI + CGRP Ab: ION-CCI rats receiving an anti-CGRP antibody. (B) The time course of the head-withdrawal threshold in the ION-CCI + CGRP Ab rats. Data were analyzed using a Friedmann's test followed by a Wilcoxon's signed rank sum test as a multiple comparison. N = 6. Significant increases were found at 1 and 2 days after the antibody administration as compared with threshold before the antibody administration. * $p < 0.05$ compared to pre. Pre: before CGRP Ab administration. 1d: at 1 day after the last administration of the anti-CGRP antibody. 2d: at 2 days after the last administration of the anti-CGRP antibody. 3d: at 3 days after the last administration of the anti-CGRP antibody.

in each group, Fig 3C). The number of pERK-IR cells was significantly increased in the ION-CCI, ION-CCI+vehi, ION-CCI+CGRP, and ION-CCI+ctrl IgG groups as compared to the sham group ($p < 0.001$ each, Fig 3C, S1 File in S1 Data). The pERK-IR cells were fewer in the ION-CCI+CGRP Ab group than in the ION-CCI, ION-CCI+vehi, ION-CCI+CGRP,

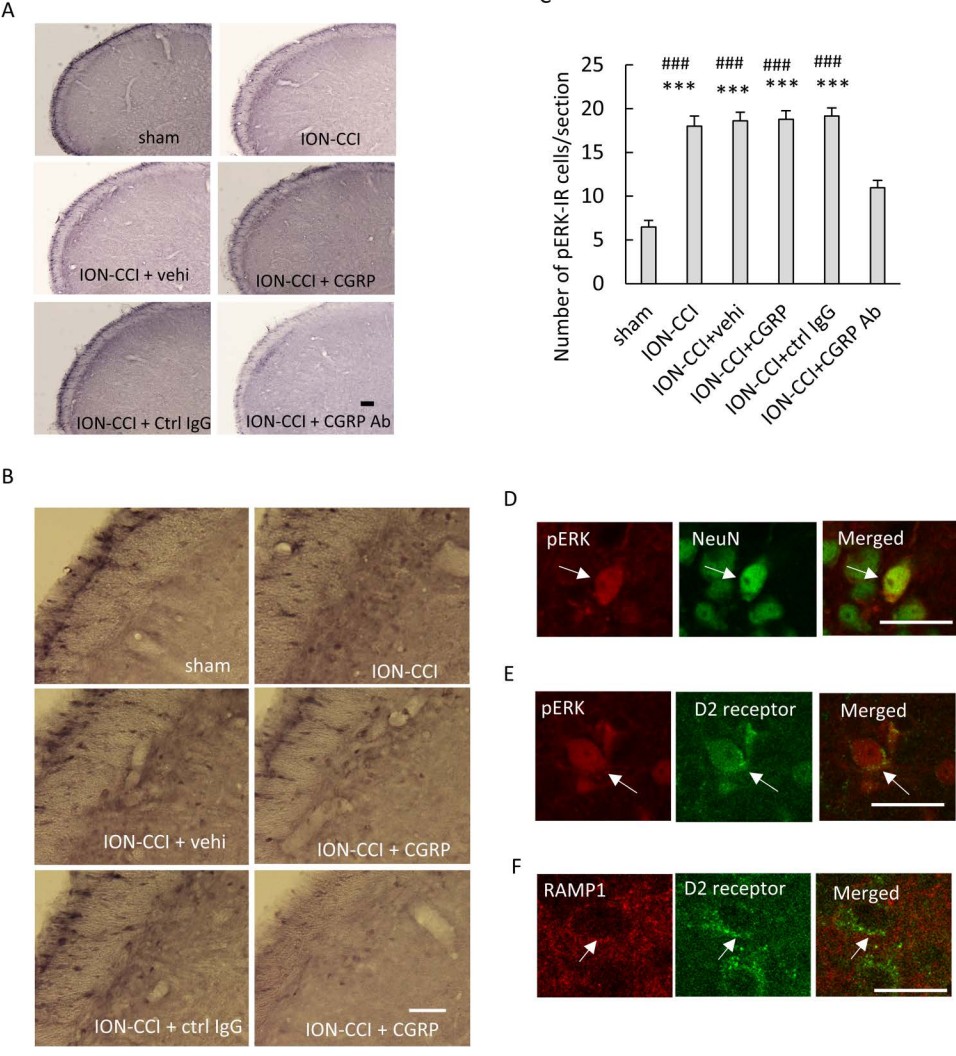

**Fig 3. Immunohistochemical reactivities to mechanical stimulation after administering drugs into the trigeminal ganglion (TG).** (A) Photomicrographs of phosphorylated extracellular signal-regulated kinase (pERK) -immunoreactive (-IR) cells in the trigeminal spinal subnucleus caudalis (Vc). Scale bar = 100 μm. (B) Photomicrographs of the pERK-IR cells with high magnification. Scale bar = 100 μm. (C) The number of pERK-IR cells in the Vc per section. Data were evaluated via a one-way analysis of variance (ANOVA) and a Tukey's post hoc test for multiple comparisons. N = 6 in each group. The number of pERK-IR cells was significantly increased in the ION-CCI, ION-CCI+vehi, ION-CCI+CGRP, and ION-CCI+ctrl IgG groups as compared to the sham group. The number of pERK-IR cells was fewer in the ION-CCI+CGRP Ab group than in the ION-CCI, ION-CCI+vehi, ION-CCI+CGRP, and ION-CCI+ctrl IgG groups. *** $p < 0.001$ compared to sham group. ### $p < 0.001$ compared to the ION-CCI+CGRP Ab group. (D) Photomicrographs of the pERK-IR and neuronal nuclei (NeuN)-IR cells, and a merged image. Colocalization of pERK and NeuN was found. Scale bar = 100 μm. (E) Photomicrographs of the pERK-IR cells, dopamine D2 receptor (D2 receptor), and a merged image. Colocalization of pERK and D2 receptor was found. Scale bar = 100 μm. (F) Photomicrograph of receptor activity modifying protein 1 (RAMP1), D2 receptor, and a merged image. Colocalization of RAMP1 and D2 receptor was found. Scale bar = 100 μm. Sham: sham-operated rats; ION-CCI: rats with chronic constriction injury of the infraorbital nerve; ION-CCI+vehi: ION-CCI rats receiving a vehicle; ION-CCI+CGRP: ION-CCI rats receiving calcitonin gene-related peptide; ION-CCI+ctrl IgG: ION-CCI rats receiving control IgG; ION-CCI+CGRP Ab: ION-CCI rats receiving an anti-CGRP antibody.

and ION-CCI+ctrl IgG groups ($p < 0.001$ each, Fig 3C). A decrease in the head-withdrawal threshold and an increase in the number of pERK-IR cells, which is defined as mechanical hypersensitivity, were observed following ION ligation in the current study, similar to the findings observed in previous studies [5,11,16]. Additionally, an increase in head-withdrawal threshold and a decrease in the number of pERK-IR cells were found after the anti-CGRP administration into the TG. Colocalization of pERK and NeuN was observed in all experimental groups following double staining of pERK and NeuN (Fig 3D, S13-S15 Figs in S1 Data). Colocalization of pERK and D2 receptor was observed following double staining of pERK and D2 receptor (Fig 3E, S16-S18 Figs in S1 Data), and the colocalization was observed in all experimental groups. Colocalization of RAMP1 and D2 receptor was observed following double staining of RAMP 1 and D2 receptor (Fig 3F, S19-S21 Figs in S1 Data), and the colocalization was also observed in all experimental groups.

### Effect of administering drugs on the ratio of CGRP-IR TG neurons

The CGRP-IR neurons were observed in the V1/V2 branch region in the TG (Fig 4A, S22-S27 Figs in S1 Data). A significant main effect was found in the ratio of the CGRP-IR TG neurons (F [5, 102] = 13.486, $p < 0.001$, n=6 in each group, Fig 4B). Significant increases in the ratio were found in the ION-CCI, ION-CCI+vehi, ION-CCI+CGRP, and ION-CCI+ctrl IgG groups as compared to sham group ($p < 0.001$, Fig 4B, S1 File in S1 Data). The ION-CCI+CGRP Ab group showed a remarkable decline as compared to the ION-CCI ($p < 0.01$), ION-CCI+vehi ($p < 0.05$), ION-CCI+CGRP ($p < 0.01$), and ION-CCI+ctrl IgG ($p < 0.05$, Fig 4B) groups. When the ratio of the number of CGRP-IR TG neurons divided by area to the total number of CGRP-IR TG neurons was analyzed, a significant difference was observed ($p < 0.001$, Fig 4C). Significant differences were found in the ION-CCI, ION-CCI+vehi, ION-CCI+CGRP, and ION-CCI+ctrl IgG groups as compared to sham group ($p < 0.001$ each, Fig 4C). A significant difference in the ratio was found between the ION-CCI+vehi and ION-CCI+CGRP Ab groups ($p < 0.001$, Fig 4C) and between the ION-CCI+ctrl IgG and ION-CCI+CGRP Ab groups ($p < 0.01$, Fig 4C).

The GFAP-IR cells were observed around the TG neurons, which is consistent with the situation where the satellite glial cells surround the neurons (Fig 5A, S28-S33 Figs in S1 Data). The ANOVA revealed a significant difference (F [5,84] = 10.512, $p < 0.001$, n=5 in each group, Fig 5B). Significant increases in the ratio of TG neurons encircled with GFAP-IR cells were found in the ION-CCI, ION-CCI+vehi, ION-CCI+CGRP, and ION-CCI+ctrl IgG groups as compared to the sham group ($p < 0.001$ each, Fig 5B). A significant decrease was found in the ION-CCI+CGRP Ab group as compared to the ION-CCI ($p < 0.05$), ION-CCI+vehi ($p < 0.05$), ION-CCI+CGRP ($p < 0.01$), and ION-CCI+ctrl IgG ($p < 0.01$, Fig 5B) groups.

### Effect of administering drugs into the TG on CGRP immunoreactivity in the Vc

CGRP immunoreactivity was observed in the superficial layers of the Vc (Fig 6A, S34-S39 Figs in S1 Data). A significant main effect was found on it (F [5, 138] = 18.695, $p < 0.001$, n=6 in each group, Fig 6B). Significant enhancement was found in the ION-CCI, ION-CCI+vehi, ION-CCI+CGRP, and ION-CCI+ctrl IgG groups as compared to the sham group ($p < 0.001$ each, Fig 6B, S1 File in S1 Data). This increase in immunoreactivity following ION ligation was consistent with the findings in a previous study [5]. The ION-CCI+CGRP Ab group revealed a remarkable decrease as compared to the ION-CCI ($p < 0.01$), ION-CCI+vehi ($p < 0.01$), ION-CCI+CGRP ($p < 0.001$), and ION-CCI+ctrl IgG ($p < 0.01$, Fig 6B) groups. Both CGRP and D2 receptor immunoreactivities were found in the superficial layers of the Vc (Fig 6C upper row, S40-S42 Figs in S1 Data). The magnified photomicrographs showed that they were in close proximity (Fig 6C lower row, S43-S45 Figs in S1 Data).

### Effect of administering drugs into the TG on the immunoreactivity of D2 receptor in the Vc

Immunoreactivity of the D2 receptor was detected primarily in the superficial layers of the Vc. Photomicrographs of D2 receptor immunoreactivity in the Vc were shown in Fig 7A (S46-S51 Figs in S1 Data), and photomicrographs of these with high magnification were shown in Fig 7B (S52-S57 Figs in S1 Data). A significant main effect was detected on

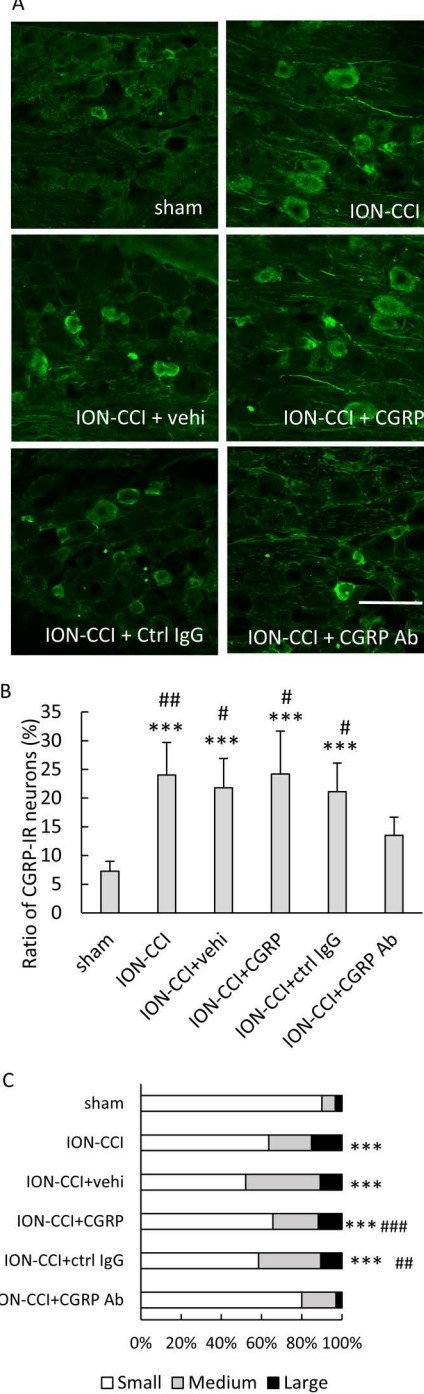

**Fig 4. Effects of administering drugs into the TG on the ratio of CGRP-IR TG neurons.** (A) CGRP-IR TG neurons in the ophthalmic (V1)/maxillary (V2) branch region. Scale bar = 100 μm. (B) The ratio of CGRP-IR TG neurons. Data were evaluated by performing a one-way ANOVA and a Tukey's post hoc test for multiple comparisons. N = 6 in each group. Significant increases were found in the ION-CCI, ION-CCI + vehi, ION-CCI + CGRP, and ION-CCI + ctrl IgG groups as compared to the sham group. The ION-CCI + CGRP Ab group revealed a remarkable decline as compared to the ION-CCI, ION-CCI + vehi, ION-CCI + CGRP, and ION-CCI + ctrl IgG groups. *** $p < 0.001$ compared to the sham group, ## $p < 0.01$, and # $p < 0.05$ compared to the ION-CCI + CGRP Ab group. (C) The ratio of CGRP-IR TG neurons divided by area to the total number of CGRP-IR TG neurons. Data were analyzed using a chi-square test with Bonferroni correction. N = 6 in each group. Significant differences were found in the ION-CCI, ION-CCI + vehi, ION-CCI + CGRP, and ION-CCI + ctrl IgG groups as compared to the sham group. A significant difference was found between the ION-CCI + vehi and

ION-CCI + CGRP Ab groups and between the ION-CCI + ctrl IgG and ION-CCI + CGRP Ab groups. *** $p < 0.001$ compared to the sham group. ### $p < 0.001$, ## $p < 0.01$ compared to ION-CCI + CGRP Ab group. Small: < 400 µm$^2$, medium: 400-800 µm$^2$, and large: > 800 µm$^2$. Sham: sham-operated rats; ION-CCI: rats with chronic constriction injury of the infraorbital nerve; ION-CCI + vehi: ION-CCI rats receiving a vehicle; ION-CCI + CGRP: ION-CCI rats receiving calcitonin gene-related peptide; ION-CCI + ctrl IgG: ION-CCI rats receiving control IgG; ION-CCI + CGRP Ab: ION-CCI rats receiving an anti-CGRP antibody.

A

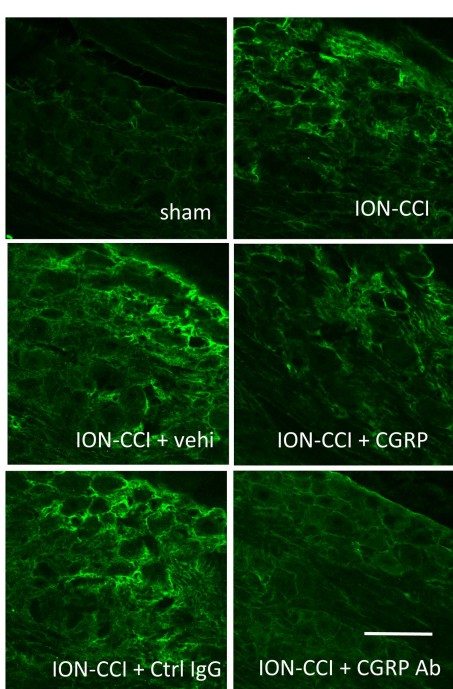

B

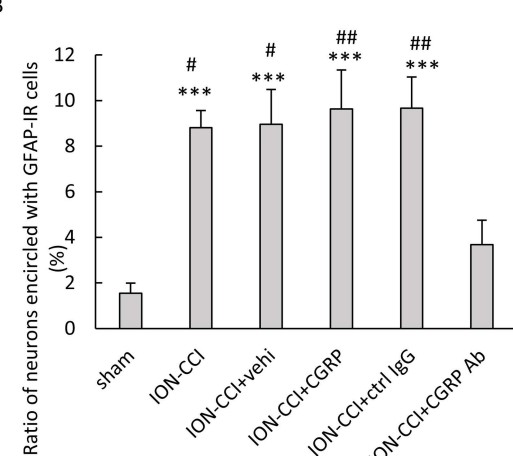

**Fig 5. Effects of administering drugs into the TG on the glial fibrillary acidic protein (GFAP)-IR cells.** (A) Photomicrographs of GFAP-IR cells. Scale bar = 100 µm. (B) The ratio of neurons encircled with GFAP-IR cells. Data were analyzed using a one-way ANOVA and a Tukey's post hoc test for multiple comparisons. N = 5 in each group. Significant increases were found in the ION-CCI, ION-CCI + vehi, ION-CCI + CGRP, and ION-CCI + ctrl IgG groups as compared to the sham group. A significant decrease was found in the ION-CCI + CGRP Ab group as compared to the ION-CCI, ION-CCI + vehi, ION-CCI + CGRP, and ION-CCI + ctrl IgG groups. *** $p < 0.001$ as compared to the sham group. ## $p < 0.01$, # $p < 0.05$ as compared to the ION-CCI + CGRP Ab group. Sham: sham-operated rats; ION-CCI: rats with chronic constriction injury of the infraorbital nerve; ION-CCI + vehi: ION-CCI rats receiving a vehicle; ION-CCI + CGRP: ION-CCI rats receiving calcitonin gene-related peptide; ION-CCI + ctrl IgG: ION-CCI rats receiving control IgG; ION-CCI + CGRP Ab: ION-CCI rats receiving an anti-CGRP antibody.

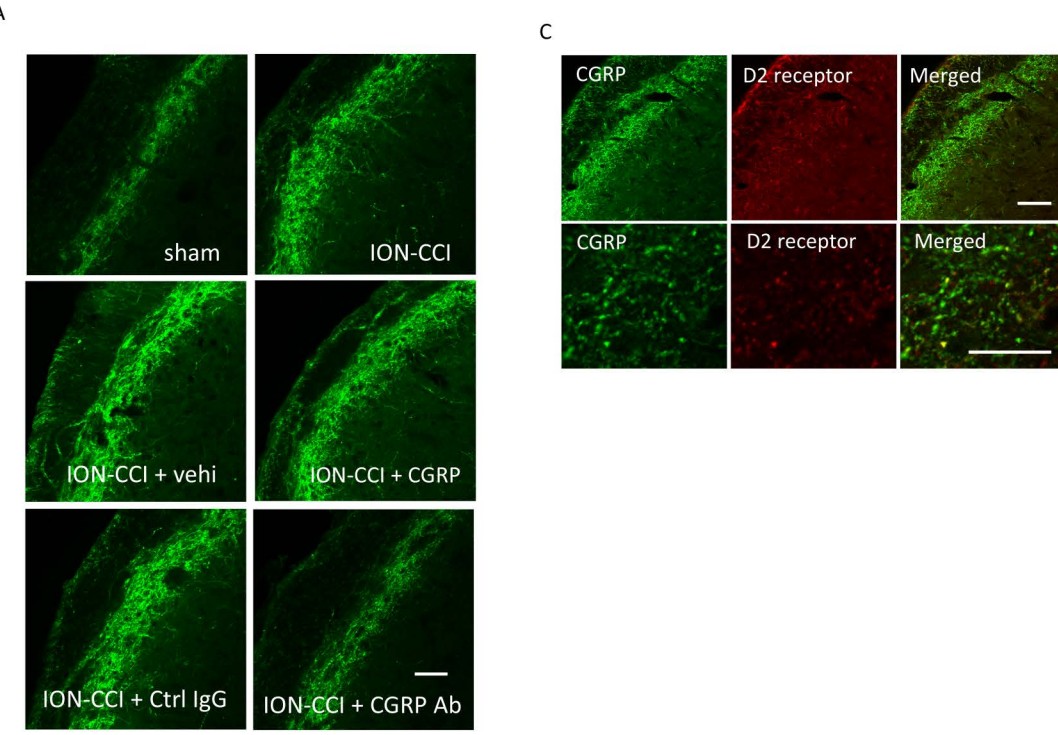

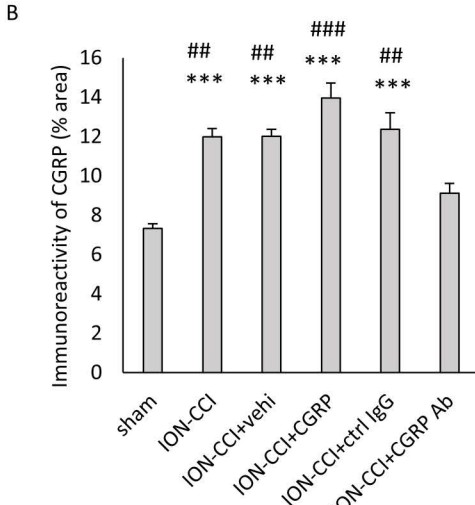

**Fig 6. Effects of administering drugs into the TG on CGRP immunoreactivity in the Vc.** (A) Immunofluorescent images of CGRP in the Vc. Scale bar = 100 µm. (B) CGRP immunoreactivity in the Vc. Data were analyzed using one-way ANOVA and a Tukey's post hoc test for multiple comparisons. N = 6 in each group. *** $p < 0.001$ as compared to the sham group, ### $p < 0.001$, and ## $p < 0.01$ as compared to the ION-CCI + CGRP Ab group. (C) Photomicrographs of CGRP, D2 receptor, and a merged image. Upper row: low magnification. Lower row: high magnification. Scale bar = 100 µm. Sham: sham-operated rats; ION-CCI: rats with chronic constriction injury of the infraorbital nerve; ION-CCI + vehi: ION-CCI rats receiving a vehicle; ION-CCI + CGRP: ION-CCI rats receiving calcitonin gene-related peptide; ION-CCI + ctrl IgG: ION-CCI rats receiving control IgG; ION-CCI + CGRP Ab: ION-CCI rats receiving an anti-CGRP antibody.

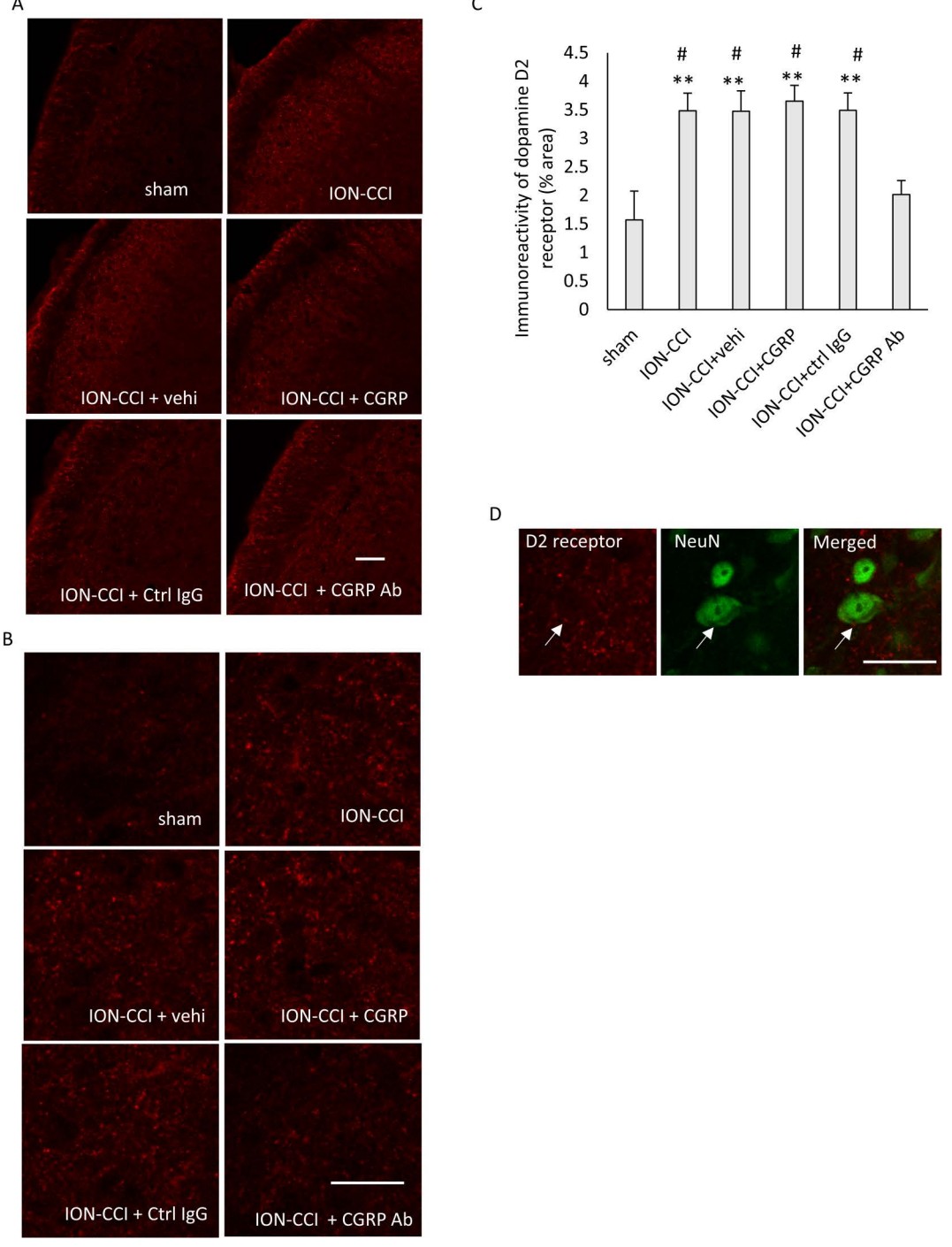

**Fig 7. Effects of administering drugs into the TG on the immunoreactivity of the D2 receptor in the Vc.** (A) Immunofluorescent images of the D2 receptor in the Vc. Scale bar = 100 μm. (B) Immunofluorescent images of the D2 receptor in the Vc with high magnification. Scale bar = 100 μm. (C) Immunoreactivity of the D2 receptor in the Vc. Data were analyzed using one-way ANOVA and a Tukey's post hoc test for multiple comparisons. N = 6 in each group. Significant increases were found in the ION-CCI, ION-CCI + vehi, ION-CCI + CGRP, and ION-CCI + ctrl IgG groups as compared to the sham group. A remarkable reduction was detected in the ION-CCI + CGRP Ab group as compared to the ION-CCI, ION-CCI + vehi, ION-CCI + CGRP, and ION-CCI + ctrl IgG groups. ** $p < 0.01$ compared to the sham group. # $p < 0.05$ compared to the ION-CCI + CGRP Ab group. (D) Photomicrographs of D2 receptor, NeuN, and a merged image, showing colocalization of the D2 receptor and NeuN. Scale bar = 100 μm. Sham: sham-operated rats; ION-CCI: rats with chronic constriction injury of the infraorbital nerve; ION-CCI + vehi: ION-CCI rats receiving a vehicle; ION-CCI + CGRP: ION-CCI rats receiving

calcitonin gene-related peptide; ION-CCI+ctrl IgG: ION-CCI rats receiving control IgG; ION-CCI+CGRP Ab: ION-CCI rats receiving an anti-CGRP antibody. NeuN: neuronal nuclei.

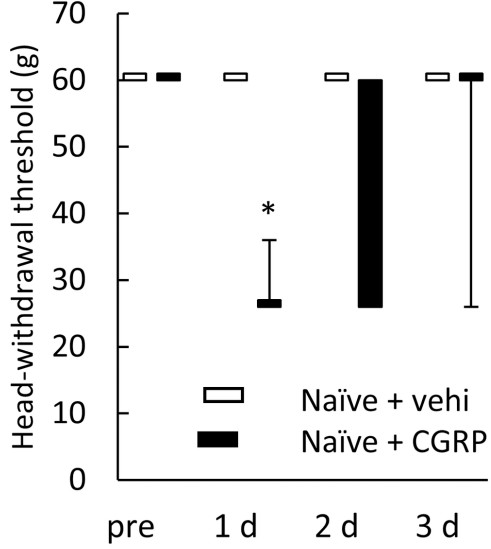

**Fig 8. The head-withdrawal threshold of naïve rats receiving CGRP.** Data were analyzed using a Friedmann's test followed by Wilcoxon's signed rank sum test as a multiple comparison to compare the threshold within the group, and a Mann–Whitney test was used for the comparison of data obtained at the same time point between the naïve rats receiving vehicle and naïve rats receiving CGRP. N = 5 in each group. In naïve rats receiving CGRP, a significant decrease was found at 1 day after the CGRP administration as compared to that before administration. * $p < 0.05$. Pre: before administration. 1 d: at 1 day after the administration. 2 d: at 2 days after the administration. 3 d: at 3 days after the administration. Vehi: vehicle. CGRP: calcitonin gene-related peptide.

immunoreactivity (F [5, 138] = 7.001, $p < 0.001$, n = 6 in each group, Fig 7C). Significant increases were found in the ION-CCI, ION-CCI+vehi, ION-CCI+CGRP, and ION-CCI+ctrl IgG groups as compared to the sham group ($p < 0.01$ each, Fig 7C, S1 File in S1 Data). A remarkable reduction was detected in the ION-CCI+CGRP Ab group as compared to the ION-CCI, ION-CCI+vehi, ION-CCI+CGRP, and ION-CCI+ctrl IgG groups ($p < 0.05$ each, Fig 7C). Double staining for D2 receptor and NeuN was performed, and colocalization of the D2 receptor and NeuN was observed (Fig 7D, S58-S60 Figs in S1 Data), which was noted in all of the experimental groups.

### Effect of CGRP administration to naïve rats on head-withdrawal threshold

Friedmann's test revealed a significant difference in head-withdrawal threshold in naïve rats receiving CGRP ($p < 0.05$, n = 5 in each group, Fig 8). In naïve rats receiving CGRP, a significant decrease was found at 1 day after CGRP administration as compared to that before administration ($p < 0.05$).

### Effects of intracisternal drug administration on the immunoreactivity of D2 receptor in the Vc

The immunoreactivity of the D2 receptor was detected primarily in the superficial layers of the Vc, similar to that shown in Fig 7 (Fig 9A, S61-S65 Figs in S1 Data). A significant difference was found on the immunoreactivity of D2 receptor (F [4,95] = 4.289, $p < 0.01$, n = 5 in each group, Fig 9B). A significant decrease was found in the ION-CCI+CGRP Ab group as compared to the ION-CCI ($p < 0.05$), ION-CCI+vehi ($p < 0.01$), ION-CCI+CGRP ($p < 0.05$), and ION-CCI+ctrl IgG ($p < 0.05$) groups.

A

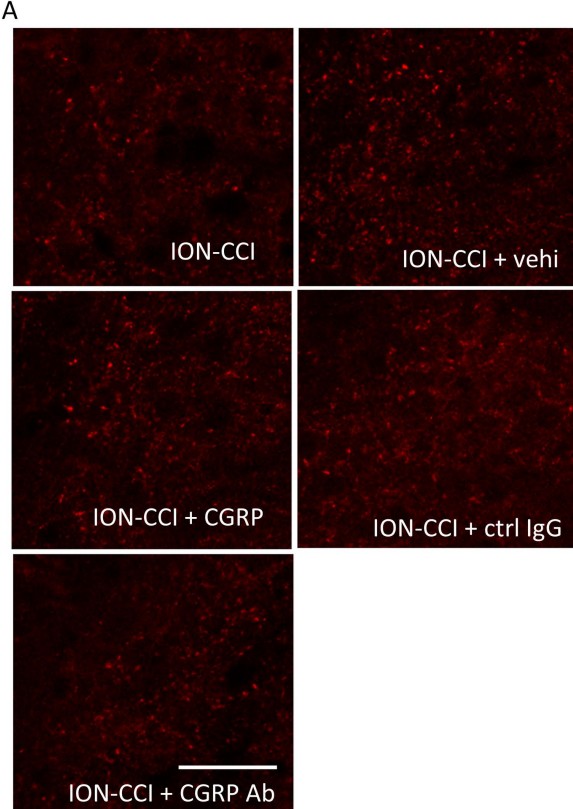

B

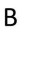
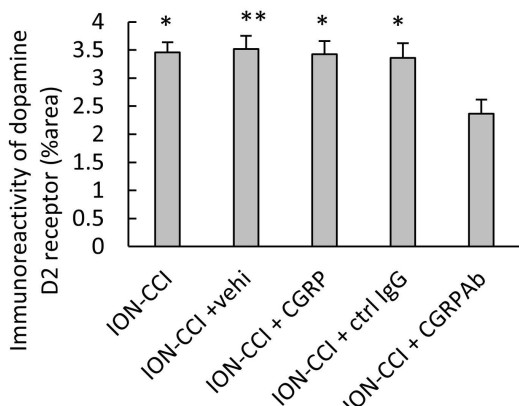

**Fig 9. Effects of intracisternal drug administration on the immunoreactivity of D2 receptor.** (A) Photomicrographs of D2 receptor in the superficial layers of the Vc. Scale bar = 100 μm. (B) Immunoreactivity of the D2 receptor. Data were analyzed using one-way ANOVA and a Tukey's post hoc test for multiple comparisons. N = 5 in each group. A significant decrease was found in the ION-CCI + CGRP Ab group as compared to the ION-CCI, ION-CCI + vehi, ION-CCI + CGRP, and ION-CCI + ctrl IgG groups. ** $p < 0.01$, * $p < 0.05$ compared to ION-CCI + CGRP Ab group. ION-CCI: rats with chronic constriction injury of the infraorbital nerve; ION-CCI + vehi: ION-CCI rats receiving a vehicle; ION-CCI + CGRP: ION-CCI rats receiving calcitonin gene-related peptide; ION-CCI + ctrl IgG: ION-CCI rats receiving control IgG; ION-CCI + CGRP Ab: ION-CCI rats receiving an anti-CGRP antibody.

## Discussion

In the present study, the head-withdrawal threshold improved and the number of pERK-IR cells in the Vc decreased following the administration of anti-CGRP antibody into the TG of ION-CCI rats. These results suggest that an anti-CGRP antibody can reduce the mechanical hypersensitivity caused by ION ligation. In a previous study, the mechanical hypersensitivity declined after administering a CGRP receptor antagonist into the TG of lingual nerve crush rats [3], indicating that CGRP and CGRP receptor in the TG play a role in trigeminal neuropathic pain. Our findings also suggest an involvement of CGRP in the TG on trigeminal neuropathic pain. The ratio of CGRP-containing neurons in the TG, large-sized CGRP-IR TG neurons, and TG neurons encircled with GFAP-IR cells increased following the occurrence of a trigeminal nerve injury [3,21]. Additionally, these ratios were reduced following the administration of a CGRP receptor antagonist [3]. In the present study, the ratios of CGRP-IR TG neurons, large-sized CGRP-IR TG neurons, and TG neurons encircled with GFAP-IR cells increased following the ION ligation and decreased following the administration of anti-CGRP antibody. These findings were also consistent with the results of a previous study [3], although the use of anti-CGRP antibody in the present study differed from what was used in the previous study. Moreover, our findings suggest an involvement of CGRP in the TG on trigeminal neuropathic pain. CGRP receptor antagonists bind to the CGRP receptor and inhibit the binding of CGRP [22]. Considering the results of our study and previous study using antagonists [3], the anti-CGRP antibodies bind to CGRP and may block its binding to the CGRP receptor. GFAP is a marker of activated satellite glial cells [23,24]. An increase in the ratio of TG neurons encircled with GFAP-IR cells suggest the activation of satellite glial cells, and its decrease indicate an inhibition of the activation of satellite glial cells. Therefore, in the present study, satellite glial cells may have been activated following the ION ligation, and the activation of these cells may be inhibited after the administration of anti-CGRP antibody. Satellite glial cells are activated in the TG following a trigeminal nerve injury [25]. TG neurons release substance P, CGRP, and tumor necrosis factor α, which activate the satellite glial cells [26]. CGRP released by the neurons induces the expression and release of nitric oxide (NO), cytokines, and brain-derived neurotrophic factor (BDNF) by the satellite glial cells via the CGRP receptors [27–29]. CGRP released by the neurons also engendered the expression of inducible NO synthase in cultured satellite glial cells [28,30]. NO, cytokines, and BDNF activate the TG neurons, promoting the expression of CGRP, CGRP receptor components, and purinergic receptor channels [31]. The interaction of neurons and satellite glial cells in the TG is referred to as "cross-talk". NO is involved in the onset and progression of neuropathic pain caused by peripheral nerve injury [32]. The nitrate and nitrite levels in the TG were enhanced in rats with inferior alveolar nerve transection [33]. In the present study, an anti-CGRP antibody administered into the TG bound to the CGRP released from the TG neurons and may have inhibited the binding of CGRP to its receptors. As a result, neuronal and satellite glial cell activation may have been suppressed, resulting in reduced mechanical hypersensitivity.

In the present study, the immunoreactivity of CGRP and D2 receptor in the Vc increased after ION ligation and decreased after the anti-CGRP antibody administration into the TG. These results are consistent with the findings of our previous studies, although the anti-CGRP antibody was administered into the TG in the present study and intracerebroventricularly in our previous studies [5,16]. Additionally, a decrease in the immunoreactivity of D2 receptor was observed in the ION-CCI rats receiving anti-CGRP antibody intracisternally. However, we performed Western blots for CGRP and D2 receptors, but we were unable to obtain reproducible data. This is a limitation of the present study. D2 receptor protein was also increased in the spinal cord following spinal [14] and sciatic [15] nerve ligations. Sciatic nerve ligation enhanced the immunoreactivity of CGRP and CGRP protein in the dorsal horn [34]. A reduction in the number of CGRP-containing DRG neurons and terminals by genetic ablation of the CGRPα-expressing sensory neurons suppresses CGRP and CGRP immunoreactivities [35]. These findings suggest that the changes in immunoreactivity reflect the alterations in the protein levels. Based on these results [34,35], if the increase or decrease in the immunoreactivities of CGRP and D2 receptor correspondingly reflects the increase and decrease in the expression of CGRP and D2 receptor, our findings suggest the increases and decreases in CGRP and D2 receptor after ION ligation

and after anti-CGRP antibody administration into the TG, respectively. In the present study, the CGRP and D2 receptor were in close proximity to each other in the Vc. The CGRP receptors reportedly exist in Vc neurons [36]. Additionally, we found the colocalization of D2 receptor and RAMP1, which is a component of the CGRP receptor, in the Vc. Our results also suggest that the D2 and CGRP receptors are expressed in the Vc neurons. Since CGRP receptor consists RAMP1 and CLR, RAMP1 immunoreactivity alone does not completely show the CGRP receptor. However, we could not find any commercially available anti-CLR antibodies. Therefore, this is a limitation of the present study. If Vc neurons have CGRP receptors, CGRP released from the nerve terminals of primary sensory neurons may bind to the CGRP receptor and then modulate protein expression through intracellular signaling mechanisms [10]. Therefore, in the present study, the expression of D2 receptors may have altered through the intracellular signaling mechanism of CGRP receptors in the Vc. Contrarily, it has been reported that RAMP1 was not found in the Vc neurons [37]. The N-methyl-d-aspartate (NMDA) and alpha-amino-3-hydroxy-5-methyl-4-isoxazole propionic acid (AMPA) receptors are present in the second order neurons of the Vc [38]. Sustained excitation of the primary nociceptive neurons, such as that caused by peripheral nerve injury or chronic inflammation, increases the synthesis and release of neurotransmitters, such as substance P and CGRP, from their central terminals, which enhances the excitability of the AMPA and NMDA receptors in the second-order neurons [39]. The increased excitability of the NMDA receptors leads to an influx of calcium ions into the cell, which can activate protein kinase C and affect transcription [10]. Administering an anti-CGRP antibody into the TG affected the immunoreactivity of the D2 receptors in the Vc. Therefore, it is suggested that acting on CGRP in the TG leads to the changes in the D2 receptor in the Vc. When dopamine binds to the D2 receptor, AC is inhibited, lowering the intracellular concentration of cAMP and suppressing PKA [15]. The effect of pain suppression by the dopaminergic nervous system is thought to originate at the D2 receptor [40]. Therefore, the alterations in the expression of D2 receptors can affect pain control mediated by the dopaminergic nervous system. The A11 cell group has dopamine-, CGRP-, and GABA-containing neurons; and these dopamine- and CGRP-containing neurons have projections to the Vc [41]. The dopaminergic pathways from the A11 cell group contribute to pain regulation [41,42]. A change in D2 receptors in the Vc may cause a change in the pain control mechanism by the dopaminergic projection from the A11 cell group.

The administration of CGRP into the TG exacerbated orofacial pain caused by the ligation of coil springs between the incisors and molars [43], although this could not be considered a neuropathic pain model. The intrathecal administration of CGRP to rats that underwent a sham operation of sciatic nerve ligation decreased the withdrawal threshold to mechanical stimulation [1]. Additionally, in the present study, the CGRP administration into the TG decreased the head-withdrawal threshold in naïve rats. Thus, it is suggested that CGRP administration elicits a decrease in threshold to mechanical stimulation. However, in the present study, no significant differences in the head-withdrawal threshold and pERK-IR cell count were observed between the ION-CCI and ION-CCI+CGRP groups. These findings indicate that the impact of CGRP administration into the TG on mechanical hypersensitivity caused by ION ligation remains unclear. An increase in the ratio of CGRP-IR TG neurons following ION ligation suggests an increase in CGRP in the TG and suggests that CGRP in the TG was already elevated before administration of CGRP. Therefore, the effect of CGRP administration following ION ligation may have been reduced. Furthermore, there may be an upper limit to the aggravation of mechanical hypersensitivity caused by an increase in CGRP. Therefore, further reduction in the threshold and increase in the number of pERK-IR cells may not have occurred. Moreover, no marked variations in the ratio of CGRP-IR TG neurons, ratio of TG neurons encircled with GFAP-IR cells, CGRP immunoreactivity, and the D2 receptor immunoreactivity were found between the ION-CCI and ION-CCI+CGRP groups. These findings may support the existence of an upper limit to the effectiveness of increased CGRP expressions.

Anti-CGRP antibody administered into the TG attenuated the mechanical hypersensitivity induced by ION ligation and decreased in the immunoreactivity of D2 receptor in the Vc. The present study suggests that the site of action of anti-CGRP antibody is TG on the suppression of mechanical hypersensitivity induced by ION ligation.

## Supporting information

**S1 Data.** **S1 Fig.** Photomicrograph of phosphorylated extracellular signal-regulated kinase (pERK)-immunoreactive (-IR) neurons in the trigeminal spinal subnucleus caudalis (Vc) of a sham rat. **S2 Fig.** Photomicrograph of pERK-IR neurons in the Vc of a chronic constriction injury of the infraorbital nerve (ION-CCI) rat. **S3 Fig.** Photomicrograph of pERK-IR neurons in the Vc of an ION-CCI rat receiving a vehicle. **S4 Fig.** Photomicrograph of pERK-IR neurons in the Vc of an ION-CCI rat receiving calcitonin gene-related peptide (CGRP). **S5 Fig.** Photomicrograph of pERK-IR neurons in the Vc of an ION-CCI rat receiving control immunoglobulin G (IgG). **S6 Fig.** Photomicrograph of pERK-IR neurons in the Vc of an ION-CCI rat receiving an anti-CGRP antibody. **S7 Fig.** High magnification photomicrograph of pERK-IR neurons in the Vc of a sham rat. **S8 Fig.** High magnification photomicrograph of pERK-IR neurons in the Vc of an ION-CCI rat. **S9 Fig.** High magnification photomicrograph of pERK-IR neurons in the Vc of an ION-CCI rat receiving a vehicle. **S10 Fig.** High magnification photomicrograph of pERK-IR neurons in the Vc of an ION-CCI rat receiving CGRP. **S11 Fig.** High magnification photomicrograph of pERK-IR neurons in the Vc of an ION-CCI rat receiving control IgG. **S12 Fig.** High magnification photomicrograph of pERK-IR neurons in the Vc of an ION-CCI rat receiving an anti-CGRP antibody. **S13 Fig.** Immunofluorescent image of pERK in the Vc. **S14 Fig.** Immunofluorescent image of neuronal nuclei (NeuN) in the Vc. **S15 Fig.** Merged image of pERK and NeuN in the Vc. **S16 Fig.** Immunofluorescent image of pERK in the Vc. **S17 Fig.** Immunofluorescent image of dopamineD2 receptor (D2 receptor) in the Vc. **S18 Fig.** Merged image of pERK and D2 receptor in the Vc. **S19 Fig.** Immunofluorescent image of receptor activity modifying protein 1 (RAMP1) in the Vc. **S20 Fig.** Immunofluorescent image of D2 receptor in the Vc. **S21 Fig.** Merged image of RAMP1 and D2 receptor in the Vc. **S22 Fig.** Photomicrograph of CGRP-IR neurons in the trigeminal ganglion (TG) of a sham rat. **S23 Fig.** Photomicrograph of CGRP-IR neurons in the TG of an ION-CCI rat. **S24 Fig.** Photomicrograph of CGRP-IR neurons in the TG of an ION-CCI rat receiving vehicle. **S25 Fig.** Photomicrograph of CGRP-IR neurons in the TG of an ION-CCI rat receiving CGRP. **S26 Fig.** Photomicrograph of CGRP-IR neurons in the TG of an ION-CCI rat receiving control IgG. **S27 Fig.** Photomicrograph of CGRP-IR neurons in the TG of an ION-CCI rat receiving an anti-CGRP antibody. **S28 Fig.** Immunofluorescent image of glial fibrillary acidic protein (GFAP) in the TG of a sham rat. **S29 Fig.** Immunofluorescent image of GFAP in the TG of an ION-CCI rat. **S30 Fig.** Immunofluorescent image of GFAP in the TG of an ION-CCI rat receiving a vehicle. **S31 Fig.** Immunofluorescent image of GFAP in the TG of an ION-CCI rat receiving CGRP. **S32 Fig.** Immunofluorescent image of GFAP in the TG of an ION-CCI rat receiving control IgG. **S33 Fig.** Immunofluorescent image of GFAP in the TG of an ION-CCI rat receiving an anti-CGRP antibody. **S34 Fig.** Immunofluorescent image of CGRP in the Vc of a sham rat. **S35 Fig.** Immunofluorescent image of CGRP in the Vc of an ION-CCI rat. **S36 Fig.** Immunofluorescent image of CGRP in the Vc of an ION-CCI rat receiving a vehicle. **S37 Fig.** Immunofluorescent image of CGRP in the Vc of an ION-CCI rat receiving CGRP. **S38 Fig.** Immunofluorescent image of CGRP in the Vc of an ION-CCI rat receiving control IgG. **S39 Fig.** Immunofluorescent image of CGRP in the Vc of an ION-CCI rat receiving an anti-CGRP antibody. **S40 Fig.** Immunofluorescent image of CGRP in the Vc. **S41 Fig.** Immunofluorescent image of D2 receptor in the Vc. **S42 Fig.** Merged image of CGRP and D2 receptor in the Vc. **S43 Fig.** High magnification immunofluorescent image of CGRP in the Vc. S44 Fig. High magnification immunofluorescent image of D2 receptor in the Vc. **S45 Fig**. High magnification merged image of CGRP and D2 receptor in the Vc. **S46 Fig.** Immunofluorescent image of D2 receptor in the Vc of a sham rat. **S47 Fig.** Immunofluorescent image of D2 receptor in the Vc of an ION-CCI rat. **S48 Fig.** Immunofluorescent image of D2 receptor in the Vc of an ION-CCI rat receiving a vehicle. **S49 Fig.** Immunofluorescent image of D2 receptor in the Vc of an ION-CCI rat receiving CGRP. **S50 Fig.** Immunofluorescent image of D2 receptor in the Vc of an ION-CCI rat receiving control IgG. **S51 Fig**. Immunofluorescent image of D2 receptor in the Vc of an ION-CCI rat receiving an anti-CGRP antibody. **S52 Fig.** High magnification immunofluorescent image of D2 receptor in the Vc of a sham rat. **S53 Fig.** High magnification immunofluorescent image of D2 receptor in the Vc of an ION-CCI rat. **S54 Fig.** High magnification immunofluorescent image of D2 receptor in the Vc of an ION-CCI rat receiving a vehicle. **S55 Fig.** High magnification immunofluorescent image of D2 receptor in the Vc of an ION-CCI rat receiving CGRP. **S56 Fig.** High magnification immunofluorescent image of D2 receptor in the Vc of

an ION-CCI rat receiving control IgG. **S57 Fig.** High magnification immunofluorescent image of D2 receptor in the Vc of an ION-CCI rat receiving an anti-CGRP antibody. **S58 Fig.** Immunofluorescent image of D2 receptor in the Vc. **S59 Fig.** Immunofluorescent image of NeuN in the Vc. **S60 Fig.** Merged image of D2 receptor and NeuN in the Vc. **S61 Fig.** Immunofluorescent image of D2 receptor in the Vc of an ION-CCI rat. **S62 Fig.** Immunofluorescent image of D2 receptor in the Vc of an ION-CCI rat receiving a vehicle. **S63 Fig.** Immunofluorescent image of D2 receptor in the Vc of an ION-CCI rat receiving CGRP. **S64 Fig.** Immunofluorescent image of D2 receptor in the Vc of an ION-CCI rat receiving control IgG. **S65 Fig.** Immunofluorescent image of D2 receptor in the Vc of an ION-CCI rat receiving an anti-CGRP antibody. **S1 File.** Raw data of behavioral and immunohistochemical responses.
(ZIP)

## Acknowledgments

The authors thank Enago ([www.enago.jp](www.enago.jp)) for English language editing.

## Author contributions

**Conceptualization:** Hiroharu Maegawa, Hitoshi Niwa.

**Data curation:** Hiroharu Maegawa, Nayuka Usami, Chiho Kudo.

**Formal analysis:** Hiroharu Maegawa, Nayuka Usami, Chiho Kudo.

**Funding acquisition:** Hiroharu Maegawa.

**Investigation:** Hiroharu Maegawa, Nayuka Usami, Chiho Kudo.

**Methodology:** Hiroharu Maegawa, Nayuka Usami.

**Project administration:** Hiroharu Maegawa.

**Resources:** Hiroharu Maegawa.

**Supervision:** Hiroharu Maegawa, Hitoshi Niwa.

**Validation:** Hiroharu Maegawa.

**Visualization:** Hiroharu Maegawa.

**Writing – original draft:** Hiroharu Maegawa.

**Writing – review & editing:** Hiroharu Maegawa, Nayuka Usami, Chiho Kudo, Hitoshi Niwa.

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
