## [Decision Letter · Decision Letter 0]

4 Dec 2024

PONE-D-24-44553Suppression of mechanical hypersensitivity and change in the expression of the dopamine D2 receptor by administration of anti-CGRP antibody into the trigeminal ganglion in trigeminal neuropathic pain model ratsPLOS ONE

Dear Dr. Maegawa,

Thank you for submitting your manuscript to PLOS ONE. After careful consideration, we feel that it has merit but does not fully meet PLOS ONE’s publication criteria as it currently stands. Therefore, we invite you to submit a revised version of the manuscript that addresses the points raised during the review process.

We look forward to receiving your revised manuscript.

Kind regards,

Kofi Asiedu, O.D., Ph.D.

Academic Editor

PLOS ONE

**Journal Requirements:**

This research was funded by JSPS KAKENHI, grant number 22K10167

Reviewers' comments:

Reviewer's Responses to Questions

**Comments to the Author**

1. Is the manuscript technically sound, and do the data support the conclusions?

Reviewer #1: Yes

Reviewer #2: Partly

Reviewer #3: Partly

Reviewer #4: Yes

2. Has the statistical analysis been performed appropriately and rigorously? 

Reviewer #1: Yes

Reviewer #2: Yes

Reviewer #3: Yes

Reviewer #4: Yes

3. Have the authors made all data underlying the findings in their manuscript fully available?

Reviewer #1: Yes

Reviewer #2: Yes

Reviewer #3: Yes

Reviewer #4: Yes

4. Is the manuscript presented in an intelligible fashion and written in standard English?

Reviewer #1: No

Reviewer #2: Yes

Reviewer #3: Yes

Reviewer #4: Yes

5. Review Comments to the Author

**Reviewer #1: ** The authors Maegawa et al. have reported that the interaction of CGRP and D2 receptor in the trigeminal ganglion neurons is involved in orofacial neuropathic pain development using infraorbital nerve ligation (ION-CCI) model rats. The following are comments on this paper:

Major

1. The authors want to mention that the interaction between pERK-IR and D2 receptor expression in Vc neurons is involved in the orofacial neuropathic pain mechanism. To clarify this question, double immunostaining of CGRP and D2 receptors should be conducted in each group of animals.

2. Furthermore, did D2 receptor immunoreactivity express in neurons or non-neuronal cells? The authors need to show the double immunohistochemistry of pERK and NeuN, as well as the D2 receptor and NeuN.

3. The authors need to test if CGRP injection into TG causes mechanical hypersensitivity in naïve rats.

4. CGRP may be released from central terminals of CGRP-IR cells in Vc and act on D2 receptor-IR cells. If this is so, the authors need to show the interaction between CGRP-IR cells and D2 receptor-expressing cells.

5. Does intrathecal administration of CGRP or CGRP Ab cause D2 receptor expression or drown regulation in Vc in ION-CCI rats? The authors need to conduct this experiment.

6. Since the authors mentioned a lot about the involvement of satellite glial cells in orofacial neuropathic pain in the discussion section, the authors need to show the immunohistochemistry data of satellite cell activation in TG following ION-CCI.

Minor

1. The authors addressed the head-withdrawal threshold (HWT) in each group of animals, sham, ION-CCI, ION-CCI+vehile injection into trigeminal ganglion (TG), ION-CCI+CGRP injection into TG, ION-CCI+IgG injection into TG and ION-CCI+CGRP antibody injection into TG. It’s better to show the time course change in HWT.

2. High-magnification photomicrographs should be addressed in Fig. 1 B and Fig. 4A.

3. Some large-sized TG cells look CGRP-IR. It is necessary to analyze the cell size of CGRP-IR cells in ION-CCI tars.

4. The photomicrographs in Fig.4A are pretty low quality. Need to show higher quality photomicrographs in Fig. 4A.

5. The discussion section is awkwardly addressed in the current paper. The discussion section should be totally rewritten.

**Reviewer #2: ** The authors found an interaction between CGRP and D2R in the Vc after IONI. TG administration of anti-CGRP antibody decreased CGRP and D2R expression in the Vc. However, the authors need to address the following issues to strengthen the manuscript.

The authors noted that the fluorescence intensity correlates with the amount of CGRP and D2R, using Ref. It is quite difficult to conclude from fluorescence alone without measuring protein levels by Western blot analysis. The authors must show Western blot data.

Pentobarbital cannot be used for anesthesia due to the welfare of the animals. Use the appropriate anesthetic.

Which cells in Vc express D2R? If CGRP projected to the Vc regulates D2R expression, then D2R and pERK are likely expressed in the same cells. Double staining of D2R and pERK should be done. Describe the cellular mechanisms by which CGRP increases D2R expression. If CGRP is altering D2R expression through changes in activity in the upper brain, there is no proof of this, and further experiments should be done.

**Reviewer #3:**  PONE-D-24-44553

In the manuscript entitled “Suppression of mechanical hypersensitivity and change in the expression of the dopamine D2 receptor by administration of anti-CGRP antibody into the trigeminal ganglion in trigeminal neuropathic pain model rats” the Authors aimed to investigate the relationship between CGRP and the dopaminergic nervous system in neuropathic pain following anti-CGRP Ab treatment in the trigeminal ganglion.

The Authors should clearly define in the introduction section the differences from the previous study (DOI: 10.1016/j.bbrc.2022.05.015) in which they already investigated the relationship between the CGRP and dopaminergic pathway. Indeed it should be highlighted the novelty of the present study together with the aim of the study. The Authors stated: “The current study aims to elucidate the relationship between CGRP and the dopaminergic nervous system in neuropathic pain further.”; to achieve this purpose they only looked at the Dopamine D2 receptor immunoreactivity in the Vc following an anti-CGRP Ab treatment, which was already investigated in a previous paper from the same Authors (doi: 10.1016/j.bbrc.2022.05.015). The only difference was the site of anti-CGRP Ab administration (cerebroventricular vs TG), but the result is the same: dopamine D2 receptor immunostaining in the Vc decreased after treatment with an anti-CGRP antibody.

In the present form this study does not seem to provide additional information to the field.

Beside the abovementioned issue I have some comments about this manuscript, which is however well written.

1. To specify in the abstract that were used male rats.

2. To better explain when the animals were sacrificed, e.g. how many days or hours from the last administration, etc.. A schematic representation would be helpful (at least as supplementary material).

3. Even if references of previous papers from the same Authors (in which they used the same CGRP antibody) are provided it is not clear to me how did they define the dose of an antibody designed to be used in IHC to be injected in the rats. Are there any toxicological test related to its use in vivo?

4. To specify that the IHC staining and analysis were performed ipsilateral to the ION-CCI

5. Line 192: “As reported previously, pERK-IR cells were primarily found in the superficial layers of the Vc”, please to add references.

6. To add the sample size calculation. Which was the primary outcome?

7. To add which test was used for testing data normality.

8. To include in the figure legends the statistical test used and N.

9. Line 305: “Our findings also suggest an involvement of CGRP and CGRP receptor in the TG on trigeminal neuropathic pain”. In which way the data in this study outline an involvement of CGRP receptor? No investigation on them was achieved.

10. The discussion is redundant and speculative, since the finding of the present study do not provide further information.

**Reviewer #4: ** This is a well-designed and presented study investigating the mechanisms of trigeminal neuropathic pain. In particular the role of CGRP and anti-CGRP antibody in trigeminal neuropathic pain model in rats. The appropriate statistical methods were used and the data is presented clearly. The results support the findings.

Minor comment:

Could you please use a better resolution images for the figures? it would make viewing of the results much easier.

6. PLOS authors have the option to publish the peer review history of their article (what does this mean? ). If published, this will include your full peer review and any attached files.

**Do you want your identity to be public for this peer review?** For information about this choice, including consent withdrawal, please see our Privacy Policy .

Reviewer #1: No

Reviewer #2: No

Reviewer #3: No

Reviewer #4: **Yes: ** Olga A. Korczeniewska

---

## [Author Response · Author response to Decision Letter 1]

9 Mar 2025

Thank you for reviewing our manuscript. We would like to provide our responses to the reviewer's comments.

Reviewer #1: The authors Maegawa et al. have reported that the interaction of CGRP and D2 receptor in the trigeminal ganglion neurons is involved in orofacial neuropathic pain development using infraorbital nerve ligation (ION-CCI) model rats. The following are comments on this paper:

Major

1. The authors want to mention that the interaction between pERK-IR and D2 receptor expression in Vc neurons is involved in the orofacial neuropathic pain mechanism. To clarify this question, double immunostaining of CGRP and D2 receptors should be conducted in each group of animals.

Response

Thank you for reviewing our manuscript. The CGRP receptor consists of receptor activity modifying protein 1 (RAMP1) and calcitonin receptor-like receptor (CLR). To show the CGRP receptor by immunostaining, double staining of RAMP1 and CLR is required. However, although anti-RAMP1 antibodies are commercially available, we could not find any commercially available anti-CLR antibodies. There are reports of self-preparing anti-CLR antibodies, but we have no experience in producing antibodies ourselves and thought it would be difficult to prepare anti-CLR antibodies. Therefore, we performed double staining of RAMP1 and D2 receptors, and obtained images suggesting colocalization of RAMP1 and D2 receptor in Vc (Fig 3F). The colocalization was observed in all experimental groups. We are aware that RAMP1 immunoreactivity alone does not completely show the CGRP receptor, but we adopted this method as an experimental method that we could use. This has also been noted as a limitation in the discussion (from page13 line 231 to page 14 line 244, from page 20 line 346 to line 348, from page 21 line 364 to line 366, and from page 33 line 574 to line 579 in manuscript; from page 14 line 248 to page 15 line 264, from page 21 line 369 to line 371, from page 22 line 390 to line 392, and from page 35 line 625 to page36 line 630 in revised manuscript with track changes).

2. Furthermore, did D2 receptor immunoreactivity express in neurons or non-neuronal cells? The authors need to show the double immunohistochemistry of pERK and NeuN, as well as the D2 receptor and NeuN.

Response

We performed double staining of Vc sections for pERK and NeuN, and D2 receptor and NeuN. Colocalization of pERK and NeuN were observed, and colocalization of D2 receptor and NeuN were observed (Fig 3D, E). These colocalizations were observed in all experimental groups. We wrote the above (from page 13 line 231 to page 14 line 244, from page 20 line 342 to line343, and from page 21 line 360 to line 362, from page 26 line 463 to line 465, from page 27 line 476 to line 477 in manuscript; from page 14 line 248 to page 15 line 264, from page 21 line 364 to line 366, and from page 22 line 385 to line 387, from page 29 line 504 to line 506, from page 29 line 517 to 519 in revised manuscript with track changes). Therefore, we considered that pERK and D2 receptor are expressed in neurons.

3. The authors need to test if CGRP injection into TG causes mechanical hypersensitivity in naïve rats.

Response

CGRP was administered to the trigeminal ganglion of naïve rats, and the threshold for the escape response to mechanical stimulation was measured. A decrease in the threshold was observed one day after the final administration (Fig 8). No significant change in the threshold was observed in the vehicle-administered group. We wrote the above (from page 8 line 142 to line 143, from page 9 line 150 to line 153, and from page 27 line 483 to page 28 line 497 in manuscript; from page 9 line 153 to line 154, page 9 line 161 to page 10 line 164, and from page 30 line 525 to page 31 line 540 in revised manuscript with track changes).

4. CGRP may be released from central terminals of CGRP-IR cells in Vc and act on D2 receptor-IR cells. If this is so, the authors need to show the interaction between CGRP-IR cells and D2 receptor-expressing cells.

Response

Double staining for CGRP and D2 receptor was performed in Vc. Immunofluorescent images showed close proximity of CGRP immunoreactivity and D2 receptor immunoreactivity (Fig 6C). We wrote the above (from page 13 line 231 to page 14 line 244, from page 25 line 438 to line 441, and from page 25 line 447 to page 26 line 449 in manuscript; from page 14 line 248 to page 15 line 264, from page 27 line 478 to line 481, and page 28 line 487 to line 489 in revised manuscript with track changes).

5. Does intrathecal administration of CGRP or CGRP Ab cause D2 receptor expression or drown regulation in Vc in ION-CCI rats? The authors need to conduct this experiment.

Response

We performed an intracisternal administration of CGRP and anti-CGRP antibody to ION-CCI rats. A decrease in D2 receptor immunoreactivity was found in ION-CCI rats receiving anti-CGRP antibodies (Fig 9). We wrote the above (from page 9 line 153 to line 157, from page 11 line 186 to page 12 line 201, and from page28 line 499 to page 29 line 516 in manuscript; from page 10 line 164 to line 168, from page 12 line 199 to line 216, and page 31 line 542 to page 32 line 561 in revised manuscript with track changes).

6. Since the authors mentioned a lot about the involvement of satellite glial cells in orofacial neuropathic pain in the discussion section, the authors need to show the immunohistochemistry data of satellite cell activation in TG following ION-CCI.

Response

Immunostaining for GFAP was performed on trigeminal ganglion sections. GFAP-immunoreactive cells were observed around TG neurons, which was consistent with the presence of activated satellite glial cells around TG neurons (Fig5A). GFAP immunoreactivity was quantified by calculating the ratio of TG neurons encircled with GFAP-immunoreactive cells. Infraorbital nerve ligation increased the ratio of TG neurons encircled with GFAP immunoreactive cells (Fig 5B). No further increase was observed after CGRP administration. Administration of anti-CGRP antibody decreased the ratio of TG neurons encircled with GFAP immunoreactive cells (Fig 5B). We wrote the above (from page 2 line 31 to line33, from page 13 line 231 to page 14 line 244, and from page 23 line 408 to page 24 line 428 in manuscript; from page 2 line 32 to line 34, from page 14 line 248 to page 15 line 264, and from page 25 line 447 to page 27 line 468 in revised manuscript with track changes).

Minor

1. The authors addressed the head-withdrawal threshold (HWT) in each group of animals, sham, ION-CCI, ION-CCI+vehile injection into trigeminal ganglion (TG), ION-CCI+CGRP injection into TG, ION-CCI+IgG injection into TG and ION-CCI+CGRP antibody injection into TG. It’s better to show the time course change in HWT.

Response

When measuring the head-withdrawal threshold after drug administration, an increase in the threshold was observed in ION-CCI rats that received anti-CGRP antibody. Therefore, we investigated the change in threshold over time in this group. Compared to before administration of anti-CGRP antibody, an increase in the threshold was observed 1 and 2 days after the end of administration (Fig 2B). We wrote the above (from page 10 line 177 to line 178, from page18 line 304 to line 308, from page 18 line 321 to page 19 line 325 in manuscript; from page 11 line 189 to line 191, from page 19 line 325 to line 328, from page 20 line 342 to line 346 in revised manuscript with track changes).

2. High-magnification photomicrographs should be addressed in Fig. 1 B and Fig. 4A.

Response

We presented high-magnification photomicrographs of Fig1B and 4A. Fig 1B is now Fig 3B. Fig 4A is now Fig 7B.

3. Some large-sized TG cells look CGRP-IR. It is necessary to analyze the cell size of CGRP-IR cells in ION-CCI tars.

Response

The size of CGRP-immunoreactive cells was analyzed (Fig 4C). Following previous literature, the size of CGRP-immunoreactive cells was divided into small, medium, and large according to area, and the ratio of the number of CGRP-immunoreactive cells of each size to the total number of CGRP-immunoreactive cells was calculated. Differences in the ratio were observed in the ION-CCI group, ION-CCI+vehi group, ION-CCI+CGRP group, and ION-CCI+ctrl IgG group compared to the sham group. In addition, differences were observed in the ratio of the ION-CCI+CGRP group and ION-CCI+ctrl IgG group compared to the ION-CCI+CGRP Ab group. We wrote the above (from page 15 line 258 to line 260, from page 17 line 289 to line 291, from page 22 line 379 to line 385, from page 23 line 395 to line 400 in manuscript; from page 16 line 278 to line 281, from page 18 line 309 to line 311, from page 24 line 417 to line 423, from page 25 line 433 to line 439 in revised manuscript with track changes).

4. The photomicrographs in Fig.4A are pretty low quality. Need to show higher quality photomicrographs in Fig. 4A.

Response

The photomicrograph in Fig 4A has been updated to a higher quality version (Fig 7A).

5. The discussion section is awkwardly addressed in the current paper. The discussion section should be totally rewritten.

Response

We have revised the discussion. We have deleted content that is not relevant to the results of the present study. We have also included the results of additional experiments based on the reviewers’ comments. As a result of these revisions, we believe that we have been able to strengthen our discussion. The corrections are follows: from page 30 line 525 to page 31 line 542, from page32 line 560 to line 563, from page 32 line 572 to page 33 line 581, from page 33 line 583 to line 591, from page 34 line 604 to line 608 in manuscript; from page 32 line 571 to page 33 line 589, from page 34 line 609 to page 35 line 613, from page 35 line 623 to page 36 line 632, from page 36 line 635 to line 643, from page 38 line 680 to page 39 line 685 in revised manuscript with track changes.

Reviewer #2: The authors found an interaction between CGRP and D2R in the Vc after IONI. TG administration of anti-CGRP antibody decreased CGRP and D2R expression in the Vc. However, the authors need to address the following issues to strengthen the manuscript.

The authors noted that the fluorescence intensity correlates with the amount of CGRP and D2R, using Ref. It is quite difficult to conclude from fluorescence alone without measuring protein levels by Western blot analysis. The authors must show Western blot data.

Response

Thank you for reviewing our manuscript. We agree with the reviewer's comment. However, we performed Western blot experiments, but unfortunately, we were unable to obtain reproducible data. We have described this in the discussion section as a limitation (from page 32 line 561 to line 563 in manuscript, from page 36 line 629 to line 630 revised manuscript with track changes).

Pentobarbital cannot be used for anesthesia due to the welfare of the animals. Use the appropriate anesthetic.

Response

We agree with reviewer’s comment. We did not use pentobarbital. We used a saline solution containing midazolam, medetomidine, and butorphanol as anesthetic in all experiments in the present study. We have mentioned the above (from page 6 line 95 to line 99 in manuscript, from page 6 line 104 to line 108 in revised manuscript with track changes).

Which cells in Vc express D2R? If CGRP projected to the Vc regulates D2R expression, then D2R and pERK are likely expressed in the same cells. Double staining of D2R and pERK should be done. Describe the cellular mechanisms by which CGRP increases D2R expression. If CGRP is altering D2R expression through changes in activity in the upper brain, there is no proof of this, and further experiments should be done.

Response

We performed double staining for D2 receptors and pERK. As a result, we found colocalization of D2 receptor and pERK (Fig. 3D). We wrote the above (from page 13 line 231 to page 14 line 244, from page 20 line 344 to line 346, from page 21 line 362 to line 364 in manuscript; from page 14 line 248 to page 15 line 264, from page 21 line 366 to line 369, from page 22 line 385 to line 387 in revised manuscript with track changes).

Colocalization of RAMP1 and D2 receptors was observed (Fig. 3F), suggesting colocalization of CGRP receptors and D2 receptors. There have also been reports that CGRP receptors exist in Vc neurons (36). We hypothesized that after CGRP binds to the CGRP receptor, an intracellular signaling mechanism is activated, leading to increased protein expression, one of which is the D2 receptor. However, another report showed that Vc neurons did not express CGRP receptors [37]. NMDA and AMPA receptors are present in the second order neurons of the Vc [38]. Sustained excitation of primary nociceptive neurons, such as that caused by peripheral nerve injury or chronic inflammation, increases the synthesis and release of neurotransmitters such as SP and CGRP from their central terminals, which enhances the excitability of AMPA and NMDA receptors in second order neurons [39]. Increased excitability of NMDA receptors leads to an influx of calcium ions into the cell, which can activate protein kinase C and affect transcription [10]. We wrote the above (from page 32 line 572 to page 33 line 591 in manuscript, from page 35 line 623 to page 36 line 643 in revised manuscript with track changes).

This study did not examine activity in the upper brain. Therefore, the contents of page19, line 341-page 20, line 351 in the first submitted manuscript were deleted because they were not considered appropriate as a discussion of the results of this study (from page 37 line 654 to page 38 line 670).

Reviewer #3: PONE-D-24-44553

In the manuscript entitled “Suppression of mechanical hypersensitivity and change in the expression of the dopamine D2 receptor by administration of anti-CGRP antibody into the trigeminal ganglion in trigeminal neuropathic pain model rats” the Authors aimed to investigate the relationship between CGRP and the dopaminergic nervous system in neuropathic pain following anti-CGRP Ab treatment in the trigeminal ganglion.

The Authors should clearly define in the introduction section the differences from the previous study (DOI: 10.1016/j.bbrc.2022.05.015) in which they already investigated the relationship between the CGRP and dopaminergic pathway. Indeed it should be highlighted the novelty of the present study together with the aim of the study. The Authors stated: “The current study aims to elucidate the relationship between CGRP and the dopaminergic nervous system in neuropathic pain further.”; to achieve this purpose they only looked at the Dopamine D2 receptor immunoreactivity in the Vc following an anti-CGRP Ab treatment, which was already investigated in a previous paper from the same Authors (doi: 10.1016/j.bbrc.2022.05.015). The only difference was the site of anti-CGRP Ab administration (cerebroventricular vs TG), but the result is the same: dopamine D2 receptor immunostaining in the Vc decreased after treatment with an anti-CGRP antibody.

In the present form this study does not seem to provide additional information to the field.

Response

Thank you for reviewing our manuscript. In our previous study in which CGRP antibodies were administered intracerebroventricularly, it was thought that anti-CGRP antibodies may bind to CGRP released from primary neurons and inhibit CGRP binding to CGRP receptors expressed in Vc. However, CGRP receptors are also present in other sites. Because anti-CGRP antibody administration was intracerebroventricular, it was difficult to identify the site of action of the CGRP antibodies. In this study, we were able to suppress hypersensitivity to mechanical stimuli by administering CGRP antibodies to the trigeminal ganglion. Some of the results obtained are similar to previous studies. However, we believe that the novelty lies in the fact that we showed that neuropathic pain was suppressed and that changes occurred in D2 receptors by acting on CGRP in the trigeminal ganglion. In other words, we were able to clarify the site involved in the inhibition of neuropathic pain more than in previous studies.

Beside the abovementioned issue I have some comments about this manuscript, which is howeve

---

## [Decision Letter · Decision Letter 1]

15 Apr 2025

Suppression of mechanical hypersensitivity and change in the expression of the dopamine D2 receptor by administration of anti-CGRP antibody into the trigeminal ganglion in trigeminal neuropathic pain model rats

PONE-D-24-44553R1

Dear Dr. Maegawa,

We’re pleased to inform you that your manuscript has been judged scientifically suitable for publication and will be formally accepted for publication once it meets all outstanding technical requirements.

Kind regards,

Kofi Asiedu, O.D., Ph.D.

Academic Editor

PLOS ONE

Additional Editor Comments (optional):

Reviewers' comments:

Reviewer's Responses to Questions

**Comments to the Author**

1. If the authors have adequately addressed your comments raised in a previous round of review and you feel that this manuscript is now acceptable for publication, you may indicate that here to bypass the “Comments to the Author” section, enter your conflict of interest statement in the “Confidential to Editor” section, and submit your "Accept" recommendation.

Reviewer #4: All comments have been addressed

Reviewer #5: All comments have been addressed

Reviewer #6: (No Response)

2. Is the manuscript technically sound, and do the data support the conclusions?

Reviewer #4: Yes

Reviewer #5: Partly

Reviewer #6: Yes

3. Has the statistical analysis been performed appropriately and rigorously? 

Reviewer #4: Yes

Reviewer #5: No

Reviewer #6: Yes

4. Have the authors made all data underlying the findings in their manuscript fully available?

Reviewer #4: Yes

Reviewer #5: Yes

Reviewer #6: Yes

5. Is the manuscript presented in an intelligible fashion and written in standard English?

Reviewer #4: Yes

Reviewer #5: Yes

Reviewer #6: Yes

6. Review Comments to the Author

Reviewer #4: (No Response)

Reviewer #5: Introduction

1. Clear Hypothesis and Objectives:

o Issue: The introduction lacks a clear statement of the hypothesis and objectives.

o Suggestion: Clearly state the hypothesis and objectives at the beginning to help readers understand the purpose and direction of the research.

2. Redundancy:

o Issue: There is redundant information regarding the role of CGRP and its receptors.

o Suggestion: Consolidate this information to make the text more concise and focused.

Methods

3. Detailed Descriptions:

o Issue: Some methodological details are missing or insufficiently described.

Example: The exact coordinates for the stereotaxic apparatus are not specified.

Example: The type of dental acrylic used for fixing the cannula is not mentioned.

Example: The method of drug administration into the TG and the intracisternal space could be described in more detail to ensure replicability.

o Suggestion: Provide these details to ensure that other researchers can replicate the study accurately.

4. Consistency in Terminology and Units:

o Issue: Inconsistencies in the use of terminology and units.

Example: Ensure that all measurements are in the same units (e.g., grams for weight, microliters for volumes).

o Suggestion: Maintain consistency in terminology and units throughout the section.

5. Ethical Considerations:

o Issue: Brief description of measures taken to minimize animal suffering is missing.

o Suggestion: Include a brief description of any measures taken to minimize animal suffering.

Statistical Analysis

6. Clear Description of Statistical Methods:

o Issue: The statistical methods are not described in sufficient detail.

Example: The software or tools used for statistical analysis are not mentioned.

Example: The rationale for choosing specific statistical tests is not explained.

o Suggestion: Clearly describe the statistical methods used, including any software or tools, and explain the rationale for choosing specific tests.

Results

7. Data Presentation:

o Issue: The data presentation could be clearer and more concise.

Example: The results could be summarized in a more straightforward manner.

Example: The use of figures and tables could be optimized to illustrate key findings.

o Suggestion: Summarize the results in a more straightforward manner and optimize the use of figures and tables.

8. Interpretation of Results:

o Issue: The results are not sufficiently discussed in relation to the hypothesis and objectives.

o Suggestion: Provide a more thorough interpretation of the findings, linking them back to the hypothesis and objectives.

9. Linking Results to Introduction:

o Issue: The results should be more clearly linked to the introduction.

o Suggestion: Ensure that the flow from the background information to the experimental design and findings is seamless.

Discussion

10. Comprehensive Discussion:

o Issue: The discussion section should comprehensively interpret the results, compare them with existing literature, and discuss the broader implications of the findings.

o Suggestion: Expand the discussion to include comparisons with existing literature and broader implications.

11. Future Directions:

o Issue: Potential future research directions are not clearly outlined.

o Suggestion: Clearly outline potential future research directions based on the findings to help understand the broader impact and next steps for the research.

Grammar and Syntax

12. Proofreading:

o Issue: There are grammatical errors and awkward phrasing.

o Suggestion: Proofread for grammatical errors and awkward phrasing to maintain a professional tone and ensure that the text is polished.

Reviewer #6: Although I was not a reviewer of the first edition, I have reviewed the changes by the authors, and they have been adequately addressed. I think this paper brings new knowledge over intrathecal administration of an antibody, #1 it addresses differences between peripheral and central mechanism #2 antibodies does not generally cross the BBB, hence therapeutically in is interesting that the antibodies work peripherally. Regarding western blots of CGRP, this can be very difficult, due to the small size of CGRP so this is also recognized as a general problem. Injection of an antibody in the TG could interfere with immunohistochemistry, nevertheless this study showed similar effects in figure 6, (Reducha PV, Bömers JP, Edvinsson L, Haanes KA. Rodent behavior following a dural inflammation model with anti-CGRP migraine medication treatment. Front Neurol. 2023 Feb 23;14:1082176. doi: 10.3389/fneur.2023.1082176). Further, if there is a second round of revision, I would consider including that there are therapeutical CGRP-antibodies available that are used to treat migraine. You are not correct that no commercial antibodies are available for CLR, for example there is this one: https://www.alomone.com/p/anti-crlr-calcrl-extracellular-antibody/ACR-060 . But, unlike the antibodies for CGRP in IHC, which work great, the antibodies for the CGRP receptors are generally not very good, for example see here: Hendrikse ER, Rees TA, Tasma Z, Garelja ML, Siow A, Harris PWR, Pawlak JB, Caron KM, Blakeney ES, Russo AF, Sowers LP, Lutz TA, Le Foll C, Walker CS, Hay DL. Characterization of Antibodies against Receptor Activity-Modifying Protein 1 (RAMP1): A Cautionary Tale. Int J Mol Sci. 2022 Dec 16;23(24):16035. doi: 10.3390/ijms232416035. So I deem the approach in the current paper satisfactory.

7. PLOS authors have the option to publish the peer review history of their article (what does this mean? ). If published, this will include your full peer review and any attached files.

**Do you want your identity to be public for this peer review?** For information about this choice, including consent withdrawal, please see our Privacy Policy .

Reviewer #4: No

Reviewer #5: **Yes: ** Randolph Jeffrey Kwaw

Reviewer #6: No

---

## [Editor Report · Acceptance letter]

PONE-D-24-44553R1

PLOS ONE

Dear Dr. Maegawa,

I'm pleased to inform you that your manuscript has been deemed suitable for publication in PLOS ONE. Congratulations! Your manuscript is now being handed over to our production team.

Kind regards,

on behalf of

Dr. Kofi Asiedu

Academic Editor

PLOS ONE